# UNIFIED MULTI-MODAL INTERACTIVE & REACTIVE 3D MOTION GENERATION VIA RECTIFIED FLOW

**Prerit Gupta**\* **Shourya Verma**\* **Ananth Grama & Aniket Bera**
Department of Computer Science
Purdue University
{gupta596, verma198, ayg, aniketbera}@purdue.edu
https://gprerit96.github.io/dualflow-page

## ABSTRACT

Generating realistic, context-aware two-person motion conditioned on diverse modalities remains a fundamental challenge for graphics, animation and embodied AI systems. Real-world applications such as VR/AR companions, social robotics and game agents require models capable of producing coordinated interpersonal behavior while flexibly switching between interactive and reactive generation. We introduce DualFlow, the first unified and efficient framework for multi-modal two-person motion generation. DualFlow conditions 3D motion generation on diverse inputs, including text, music, and prior motion sequences. Leveraging rectified flow, it achieves deterministic straight-line sampling paths between noise and data, reducing inference time and mitigating error accumulation common in diffusion-based models. To enhance semantic grounding, DualFlow employs a novel Retrieval-Augmented Generation (RAG) module for two-person motion that retrieves motion exemplars using music features and LLM-based text decompositions of spatial relations, body movements, and rhythmic patterns. We use contrastive rectified flow objective to further sharpen alignment with conditioning signals and add synchronization loss to improve inter-person temporal coordination. Extensive evaluations across interactive, reactive, and multi-modal benchmarks demonstrate that DualFlow consistently improves motion quality, responsiveness, and semantic fidelity. DualFlow achieves state-of-the-art performance in two-person multi-modal motion generation, producing coherent, expressive, and rhythmically synchronised motion.

## 1 INTRODUCTION

Generating realistic, context-aware interactive human motion remains a core challenge in computer graphics and human–computer interaction (Holden et al., 2016). Synthesizing coordinated multi-person behavior requires capturing mutual responsiveness, physical plausibility, and interpersonal dynamics which is essential for immersive VR/AR, game AI, and human-robot collaboration. Since interactions are driven by multi-modal stimuli (language, music, physical cues), generative systems must integrate these inputs. Real-world embodied agents must also switch between interactive coordination with other agents and reactive adaptation to human partners, making flexible multi-task generation crucial. Existing two-person motion approaches treat interactive and reactive settings as separate tasks with incompatible architectures, training objectives, and conditioning signals. Interactive models (Liang et al., 2024; Ghosh et al., 2025) focus on bidirectional coordination without handling asymmetric reactive generation, while reactive models Rahman et al. (2022); Xu et al. (2024); Siyao et al. (2024) specialize in predicting reactor motion from actor cues. Current methods support only single-modality conditioning: text-only (Liang et al., 2024; Xu et al., 2024) or music-only (Siyao et al., 2024; Ghosh et al., 2025). No unified model performs both tasks under one architecture while leveraging multi-modal cues.

We introduce DualFlow, the first unified multi-modal rectified flow framework for interactive and reactive two-person motion generation (Fig. 1). DualFlow employs cascaded DualFlow Blocks that

---
\*Equal contribution.

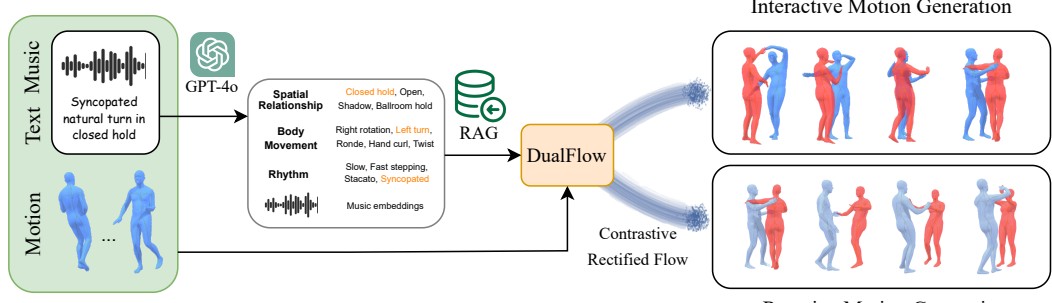

Figure 1: Our DualFlow model unifies two tasks: (a) Interactive Motion Generation, which synthesizes synchronized two-person interactions, (b) Reactive Motion Generation, which generates responsive motions for Person B (red) conditioned on Person A's (blue) movements. The generation process is conditioned jointly on text, music, and the retrieved motion samples.

adapt through masking: both branches activate for interactive generation, while only the reactor branch conditions on actor motion for reactive synthesis. This enables task switching without retraining and shared representation learning.

DualFlow incorporates a novel RAG adaptation for two-person motion. Unlike single-person RAG modules, our model retrieves semantically relevant samples using LLM-decomposed interactive text descriptions (spatial relationships, body movements, rhythm) and music features. Retrieved samples inject through retrieval-based cross-attention in each DualFlow block, grounding generation in interaction-aware exemplars and improving spatial-semantic alignment. DualFlow further employs Contrastive Rectified Flow generation, where contrastive learning sharpens motion embeddings, improves inter-person relational consistency, and strengthens motion-condition alignment. Combined with rectified flow sampling (faster convergence, reduced error accumulation), these contrastive objectives enhance diversity, coherence, and semantic fidelity.

Our key contributions are: (1) Unified architecture for interactive and reactive two-person motion generation with seamless task switching. (2) A Retrieval-Augmented Generation (RAG) framework for two-person motion generation leveraging music features and interactive text-based descriptions (spatial relationship, body movement, rhythm) decomposed using LLM to guide semantically faithful motion synthesis. (3) Contrastive Rectified Flow based generation with added synchronization loss, improving motion quality, semantic alignment and faster sampling. (4) Extensive quantitative, qualitative, and ablation studies on diverse two-person datasets, showing DualFlow generates coherent, expressive, and contextually appropriate motions with fewer sampling steps. Importantly, our approach outperforms state-of-the-art baselines by 2.5% in FID, 76% in R-precision, 3x in Multi-Modal Distance for Interactive task, 1.7% in FID, 2.5x in R-precision, 2x in Multi-Modal Distance for Reactive task on MDD Dataset requiring only 20 inference steps (2.5x faster) than 50-DDIM standard, establishing new benchmark for multi-person, multi-modal motion generation.

## 2 RELATED WORK

**Two-person Motion Generation.** While single-person motion generation has advanced rapidly (Guo et al., 2022; Tevet et al., 2022; Petrovich et al., 2022; Zhang et al., 2024), extending these methods to multi-person settings introduces the additional challenge of modeling coordination between agents. Early two-person models (Kundu et al., 2020; Xu et al., 2023; Xie et al., 2021) demonstrated feasibility but exhibited limited generalization or weak semantic grounding. To address data scarcity and modeling complexity, Liang et al. (2024) introduced a large-scale interaction dataset with a text-conditioned diffusion model, later extended by text-guided variants (Shafir et al., 2024; Yi et al., 2024; Li et al., 2024a). In the domain of dance, specialized frameworks explored music-conditioned lead–follower generation (Li et al., 2024b; Wang et al., 2025a; Ghosh et al., 2025). Despite these advances, most diffusion-based methods remain slow and restricted to single-modality conditioning. For reactive motion generation, early GAN- and transformer-based meth-

ods (Men et al., 2022; Rahman et al., 2022; Ghosh et al., 2024) have recently been extended with text (Xu et al., 2024; Cen et al., 2025) or with joint leader motion and music (Siyao et al., 2024). However, existing interactive and reactive models are developed as separate systems with incompatible architectures and training objectives, limited multi-modal support and preventing seamless switching between tasks. Our framework, DualFlow, addresses these limitations by unifying interactive and reactive two-person motion generation within a single transformer-based rectified flow architecture, jointly conditioned on text, music, and retrieved motion exemplars.

**Retrieval-Augmented Generation (RAG).** RAG has significantly improved generative fidelity across language models (Gao et al., 2023; Guu et al., 2020; Lewis et al., 2020), image synthesis (Blattmann et al., 2022; Chen et al., 2022; Sheynin et al., 2022), and video generation (He et al., 2023). Within motion generation, retrieval-based approaches have been applied to text-to-motion synthesis (Zhang et al., 2023; Kalakonda et al., 2025; Liao et al., 2024; Petrovich et al., 2023; Bensabath et al., 2024), but all existing methods operate exclusively in the single-person setting and do not address interactive multi-person dynamics. DualFlow introduces the first RAG framework for two-person motion generation, retrieving interaction-aware motion exemplars using music features and LLM-based text decompositions capturing spatial relationships, body movements, and rhythmic structure. These exemplars are integrated through a retrieval-based cross-attention mechanism providing fine-grained semantic grounding crucial for coordinated two-person motion generation.

**Diffusion and Flow-based Models.** Diffusion-based motion generation models such as MDM (Tevet et al., 2022), MotionDiffuse (Zhang et al., 2024), and MoFusion (Dabral et al., 2023) have demonstrated strong performance with fewer than a hundred denoising steps, but they remain limited to single-person generation. More recent approaches adopt Flow Matching (Lipman et al., 2023) to bypass iterative denoising (Hu et al., 2023; Canales Cuba & Gois, 2025). Yet these methods face optimization instabilities and scaling difficulties when extended to multi-person motion. Inter-Gen (Liang et al., 2024), TIMotion (Wang et al., 2025b) are diffusion models tailored for two-person generation needing roughly 50 denoising steps for inference. Our framework builds on Rectified Flow (Liu et al., 2022), which introduces a deterministic straight-line transport map between noisy and clean samples, yielding simpler training dynamics and significantly faster (20 steps), more stable sampling. DualFlow extends this paradigm with a contrastive rectified flow objective that sharpens motion representations and strengthens alignment with multi-modal conditioning signals.

# 3 METHODS

## 3.1 PROBLEM FORMULATION

A two-person motion interaction $\mathbf{x} \in \mathcal{X}_{\mathcal{A}} \times \mathcal{X}_{\mathcal{B}}$ is represented as person A's motion $\mathbf{x_a} = \{x_a^i\}_{i=1}^N$ and person B's motion $\mathbf{x_b} = \{x_b^i\}_{i=1}^N$, where paired frames $x^j = (x_a^j, x_b^j)$ are naturally synchronized. For the asymmetric case, person A is the *Actor* and person B the *Reactor*. The motion space is $\mathcal{X} \subset \mathbb{R}^{N \times J \times 3}$, with $N$ frames and $J$ joints. Music features lie in $\mathcal{M} \subset \mathbb{R}^{N \times d_m}$ with dimension $d_m$, and text embeddings in $\mathcal{C} \subset \mathbb{R}^{d_c}$ with dimension $d_c$.

**Interactive Motion Generation.** Given text $c \in \mathcal{C}$ and/or music $m \in \mathcal{M}$, the task is to generate synchronized two-person motion $(\mathbf{x_a}, \mathbf{x_b})$ aligned with both modalities: $F(c, m) \mapsto \mathbf{x}$ Special cases include text-only ($m = \phi$) (Liang et al., 2024), music-only ($c = \phi$) (Li et al., 2024b; Ghosh et al., 2025)), and joint text-music conditioning defined as Text-to-Duet by Gupta et al. (2025).

**Reactive Motion Generation.** Given the actor's motion $\mathbf{x_a} \in \mathcal{X}$, text $c \in \mathcal{C}$, and/or music $m \in \mathcal{M}$, the goal is to generate the reactor's motion $\mathbf{x_b} \in \mathcal{X}$ such that $(\mathbf{x_a}, \mathbf{x_b})$ are coherent and synchronized: $G(c, m, \mathbf{x_a}) \mapsto \mathbf{x_b}$. Variants include text-only ($m = \phi$) (Xu et al., 2024), music-only ($c = \phi$) (Siyao et al., 2024), and joint text-music conditioning defined as Text-to-Dance Accompaniment by Gupta et al. (2025).

**Human Motion Representation.** We represent motion in a global coordinate system, where the origin is anchored at the root joint of person A. The position of person B is expressed relative to this root, ensuring a unified spatial reference frame for both. Our motion representation is based on the format introduced by Liang et al. (2024), and encodes a single frame of an individual's motion as $x^i = [j_g^p, j_g^v, j^r, c^f]$. Each frame includes global joint positions $j_g^p \in \mathbb{R}^{3N_j}$, global joint velocities $j_g^v \in \mathbb{R}^{3N_j}$, local joint rotations $j^r \in \mathbb{R}^{6(N_j-1)}$ in 6D format within a root-relative coordinate frame,

and binary foot contact indicators $c^f \in \mathbb{R}^4$ that specify ground contact status for each foot joint at that time step. To model body joint rotations, we use the SMPL model (Loper et al., 2015) with $N_j = 22$ joints, resulting in a fixed input dimension of $x_i \in \mathbb{R}^{262}$.

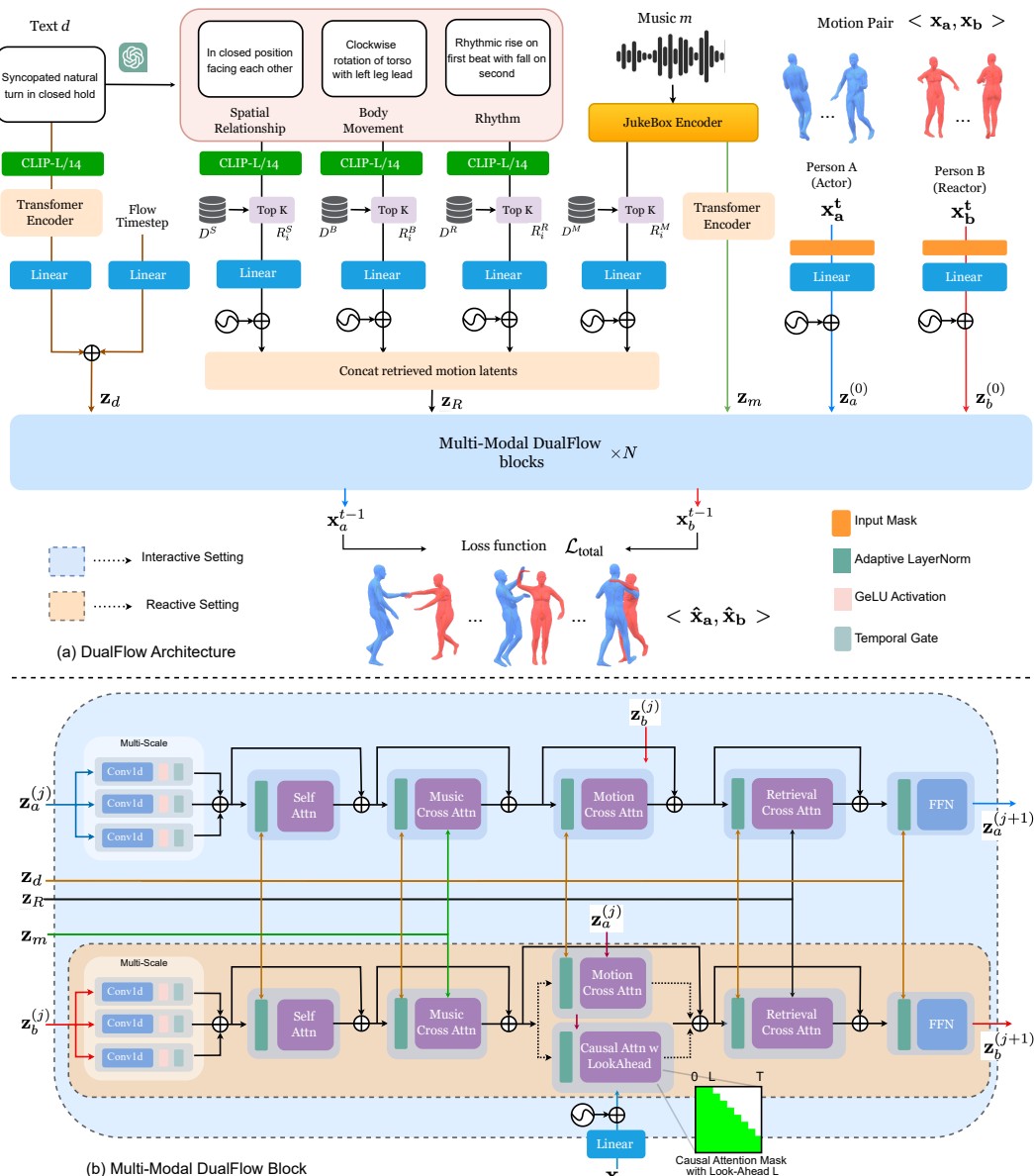

Figure 2: (a) Our framework takes text (CLIP-L/14), music, and motion sequences from an actor (A) and reactor (B) as inputs. Motion samples are retrieved using music features and LLM-decomposed text cues (spatial relationship, body movement, rhythm). These modality-specific latents are processed by cascaded Multi-Modal DualFlow Blocks that model interactive dynamics. Outputs are either both actors' motions (interactive) or only the reactor's motion (reactive) via a masking mechanism. (b) A DualFlow Block: in the interactive setting, both branches operate symmetrically with Motion Cross Attention coordinating joint motion; in the reactive setting, the actor branch is masked and the reactor branch employs a Causal Cross Attention module with Look-Ahead $L$, replacing Motion Cross Attention, to condition on the actor's motion.

## 3.2 Multi-Modal Motion Retrieval

**Retrieval Dataset.** Direct retrieval from raw text often overlooks the nuanced dimensions of interactive human motion, yielding low diversity or biased matches. To address this, we use GPT-4o to decompose each prompt into three focused descriptions (Hurst et al., 2024), inspired by Laban Movement Analysis (Laban, 1950) and aligned with the MDD Dataset (Gupta et al., 2025): **(1) Spatial Relationship** (proximity, orientation, handholds), **(2) Body Movement** (actions, body parts, posture), and **(3) Rhythm** (timing, musicality, stepping). To achieve high-quality and consistent decomposition, we design a structured prompting framework for the LLM (details in Appendix). For each category, we build retrieval databases using CLIP (Radford et al., 2021) embeddings ($D^S, D^B, D^R$) and music embeddings ($D^M$) from Jukebox (Dhariwal et al., 2020).

**Similarity Scoring.** We generalize the similarity scoring function introduced by Zhang et al. (2023) for any modality $q$. For a given query sample $p$ with modality-specific feature embedding $f_p^q$, and a candidate database motion sample $\mathbf{x}_i$ with embedding $f_i^q$ and motion length $l_i$, the similarity score $s_i^q$ is computed as:

$$s_i^q = \langle f_i^q, f_p^q \rangle \cdot e^{-\lambda \cdot \frac{|l_i - l_p|}{\max\{l_i, l_p\}}} \tag{1}$$

where $\langle \cdot, \cdot \rangle$ is cosine similarity and the exponential term penalizes mismatch across motion length with sensitivity $\lambda$, allowing retrievals that are semantically aligned and temporally compatible. Using this scoring, we retrieve top-$k$ matches from each database for every sample, yielding sets $(R_i^S, R_i^B, R_i^R, R_i^M)$ as shown in Fig. 2. The retrieved sets collectively offer a diverse yet semantically relevant collection of motion exemplars, which are later used to guide generation.

## 3.3 Model Architecture

**Conditioning latents.** The text description $d$ is encoded using a pretrained CLIP model (Radford et al., 2021) followed by a transformer encoder, then linearly projected and fused with time-step embeddings to form the text latent $\mathbf{z}_d$. Similarly, the music input $m$ is processed by a pretrained Jukebox encoder (Dhariwal et al., 2020), linearly transformed, and passed through a transformer encoder to obtain the music latent $\mathbf{z}_m$. For retrieval-based conditioning, we use four retrieved motion sets $(R_i^S, R_i^B, R_i^R, R_i^M)$ corresponding to spatial, body, rhythm, and music cues. Positional encodings and a shared linear projection map these samples to the motion latent space, and the resulting features are concatenated into the aggregated retrieval latent $\mathbf{z}_R$.

**Model Pipeline.** Motion inputs $\mathbf{x}_a^t$ and $\mathbf{x}_b^t$ sampled for time step $t$ are first projected through individual linear layers, followed by the addition of positional encodings, resulting in initial motion latents $\{\mathbf{z}_a^{(0)}, \mathbf{z}_b^{(0)}\}$. They are fed into the main pipeline consisting of $N$ cascaded DualFlow blocks. The first block takes the initial motion latents $\{\mathbf{z}_a^{(0)}, \mathbf{z}_b^{(0)}\}$ as input. Each subsequent block $(j+1)$ takes the outputs from the previous block $\{\mathbf{z}_a^{(j)}, \mathbf{z}_b^{(j)}\}$ and produces updated latents $\{\mathbf{z}_a^{(j+1)}, \mathbf{z}_b^{(j+1)}\}$, where $j \in \{0, 1, \ldots, N-1\}$. All blocks are jointly conditioned on the multi-modal context $\{\mathbf{z}_d, \mathbf{z}_m, \mathbf{z}_R\}$. The output from the last block, $\{\mathbf{x}_a^{t-1}, \mathbf{x}_b^{t-1}\}$, gives the denoised motion.

**DualFlow Block.** Each DualFlow block refines motion representations through temporally-aware and context-conditioned operations. It begins with a multi-scale temporal convolution module with varying kernel sizes to capture motion patterns at different time resolutions, followed by a GELU activation (Hendrycks & Gimpel, 2023). Branch outputs are adaptively fused using learnable gating weights $\gamma_k$. The representation then passes through a Self-Attention layer to model internal temporal dependencies, followed by a structured sequence of Cross-Attention layers: (1) *Music Cross-Attention* to align motion with music latent $\mathbf{z}_m$, (2) *Motion Cross-Attention* to capture inter-person interaction which gets replaced by *Casual Cross-Attention with Look-Ahead* during reactive setting and (3) *Retrieval Cross-Attention* to semantically guide generation using retrieved exemplars. All modules use residual connections for stability, and the text latent $\mathbf{z}_d$ is injected via LayerNorm conditioning. Each block thus integrates temporal structure, musical rhythm, and semantic guidance from retrieval. Please refer to Appendix for detailed description of each module.

**Task settings.** In interactive setting, both $\mathbf{x}_a^t$ and $\mathbf{x}_b^t$ are sampled for time step $t$ as input. In reactive setting, only reactor's motion $\mathbf{x}_b$ is sampled, while actor's motion $\mathbf{x}_l$ is masked on the input side and used for conditioning. To enable anticipatory reactor response, the *Motion Cross-attention* is switched with Causal Cross Attention Layer having a Look-Ahead parameter $L$. It uses an upper

triangular mask such that reactor's motion attends to past and only $L$ future frames of actor's motion (Fig.2). This look-ahead mechanism ensures temporally aligned and context-aware generation.

## 3.4 CONTRASTIVE RECTIFIED FLOW

To generate realistic and semantically grounded duet motions, we build upon the Rectified Flow Matching framework (Liu et al., 2022) and augment it with a contrastive learning objective inspired by Contrastive Flow Matching Stoica et al. (2025). Unlike traditional diffusion models that rely on stochastic denoising, rectified flow formulates the generation process as a deterministic Ordinary Differential Equation (ODE) that transports a noise sample toward a data sample along a straight-line path in motion space. Given a ground truth motion sample $\mathbf{x}_0$ and a noise sample $\boldsymbol{\epsilon} \sim \mathcal{N}(0, \mathbf{I})$, the interpolated state at time $t \in [0, 1]$ is defined as: $\mathbf{x}_t = (1 - t)\boldsymbol{\epsilon} + t\mathbf{x}_0$, and $\mathbf{v}_t = \mathbf{x}_0 - \boldsymbol{\epsilon}$, where $\mathbf{x}_t$ lies along the linear path from noise $\boldsymbol{\epsilon}$ at $t = 0$ to data $\mathbf{x}_0$ at $t = 1$, and $\mathbf{v}_t$ is the constant velocity vector guiding the transport. During inference, we integrate forward from $t = 0$ to $t = 1$ starting from $\boldsymbol{\epsilon} \sim \mathcal{N}(0, \mathbf{I})$ to obtain the generated motion. We train a time-dependent neural velocity field $\mathbf{v}_\theta(\mathbf{x}_t, t, c)$ to approximate $\mathbf{v}_t$, conditioned on a multimodal context $c = (d, m, R_i^S, R_i^B, R_i^R, R_i^M)$, which includes the text description $d$, music segment $m$, and retrieved motion sets. This context is encoded using cross attention layers in DualFlow Block. The flow loss $\mathcal{L}_{\text{flow}}$ is obtained by minimizing the squared error between the predicted and target velocity:

$$\mathcal{L}_{\text{flow}} = \mathbb{E}_{\mathbf{x}_0, \boldsymbol{\epsilon}, t} \left[ \| \mathbf{v}_\theta(\mathbf{x}_t, t, c) - (\mathbf{x}_0 - \boldsymbol{\epsilon}) \|_2^2 \right] \tag{2}$$

To encourage semantic alignment, we introduce a triplet contrastive loss that enforces proximity in velocity space for semantically similar prompts with $d(\cdot, \cdot)$ denoting cosine distance:

$$\mathcal{L}_{\text{triplet}} = \mathbb{E} \left[ \max \left( 0, d(\hat{\mathbf{v}}, \mathbf{v}^+) - d(\hat{\mathbf{v}}, \mathbf{v}^-) + m \right) \right] \tag{3}$$

For each batch, we randomly select an anchor sample whose predicted velocity is denoted as $\hat{\mathbf{v}} = \mathbf{v}_\theta(\mathbf{x}_t, t, c)$. We compute the cosine similarity between this anchor and all remaining samples in the batch. Positive samples $\mathbf{v}^+$ are defined as velocities belonging to motions with high semantic or structural affinity to the anchor such as those sharing the same movement style, exhibiting similar textual descriptors or aligning in rhythmic structure. Negative samples $\mathbf{v}^-$ correspond to motions that differ substantially in style or exhibit low text similarity ($> 0.6$). This sampling strategy leverages the hierarchical structure of our RAG module to construct meaningful triplets that emphasize semantically relevant distinctions. We use a margin of $m = 0.2$ and set the triplet loss weight to $\lambda_{\text{triplet}} = 0.1$. We define contrastive flow loss $\mathcal{L}_{\text{CRF}}$ that combines both losses:

$$\mathcal{L}_{\text{CRF}} = \mathcal{L}_{\text{flow}} + \lambda_{\text{triplet}} \mathcal{L}_{\text{triplet}} \tag{4}$$

Here, $\lambda_{\text{triplet}}$ balances reconstruction and semantic alignment objective.

## 3.5 REGULARIZATION LOSSES

**Geometric Losses.** We adopt the common geometric losses for human motion such as foot contact loss $\mathcal{L}_{\text{foot}}$ and joint velocity loss $\mathcal{L}_{\text{vel}}$ from MDM Tevet et al. (2022) and bone length loss $\mathcal{L}_{\text{BL}}$ from InterGen Liang et al. (2024). The geometric loss is defined as:

$$\mathcal{L}_{\text{geo}} = \mathcal{L}_{\text{foot}} + \lambda_{\text{vel}} \mathcal{L}_{\text{vel}} + \lambda_{\text{BL}} \mathcal{L}_{\text{BL}} \tag{5}$$

where the hyper-parameters $\lambda_{\text{vel}}, \lambda_{\text{BL}}$ are appropriately calibrated to fix the importance of each term.

**Interaction Losses.** We adapt joint distance map loss $\mathcal{L}_{\text{DM}}$ and relative orientation loss $\mathcal{L}_{\text{RO}}$ from InterGen Liang et al. (2024) that allows close interactions when dancers should be in contact as well as maintain proper facing directions and body alignments. To further strengthen inter-person coordination during duet generation, we introduce a synchronization loss $\mathcal{L}_{\text{sync}}$ that explicitly enforces spatial relational coherence between the two person. The loss weights pairwise inter-person joint distances using anatomically informed and task-relevant importance terms:

$$\mathcal{L}_{\text{sync}} = \sum_{j_1, j_2} w_{\text{d}}(j_1, j_2) \, w_{\text{j}}(j_1, j_2) \, \| d_{\text{p}}(j_1, j_2) - d_{\text{gt}}(j_1, j_2) \|^2, \tag{6}$$

where $d_{\text{p}}(j_1, j_2)$ and $d_{\text{gt}}(j_1, j_2)$ denote the predicted and ground-truth Euclidean distances between joint pairs across the two person. The distance-based weight $w_{\text{d}}(j_1, j_2)$ assigns higher importance

to joint pairs that are naturally closer during interaction:

$$w_{\mathrm{d}}(j_1, j_2) = e^{\left(-\alpha \left\| d_{\mathrm{gt}}(j_1,j_2) \right\|\right)}. \tag{7}$$

Complementarily, $w_{\mathrm{j}}(j_1, j_2)$ captures the anatomical & functional relevance of different body parts:

$$w_{\mathrm{j}}(j_1, j_2) = \begin{cases} \mathrm{w_h}, & \text{if } j_1, j_2 \in \mathcal{J}_{\mathrm{hands}}, \\ \mathrm{w_u}, & \text{if } j_1, j_2 \in \mathcal{J}_{\mathrm{upper}}, \\ \mathrm{w_l}, & \text{if } j_1, j_2 \in \mathcal{J}_{\mathrm{lower}}, \\ \mathrm{w_{small}}, & \text{otherwise.} \end{cases} \tag{8}$$

Here, $\mathcal{J}_{\mathrm{hands}}$ (hands, wrists), $\mathcal{J}_{\mathrm{upper}}$ (shoulders, elbows, torso), and $\mathcal{J}_{\mathrm{lower}}$ (hips, knees, feet) denote anatomically defined joint groups. Together, these weighting terms encourage the model to preserve high-frequency synchrony while maintaining the global relational structure across the two bodies.

The interaction loss $\mathcal{L}_{\mathrm{inter}}$ is obtained as:

$$\mathcal{L}_{\mathrm{inter}} = \mathcal{L}_{\mathrm{DM}} + \lambda_{\mathrm{RO}}\mathcal{L}_{\mathrm{RO}} + \lambda_{\mathrm{sync}}\mathcal{L}_{\mathrm{sync}} \tag{9}$$

where the hyper-parameters $\lambda_{\mathrm{RO}}$ and $\lambda_{\mathrm{sync}}$ are fixed based on importance of each term. For reactive setting, ground-truth actor's motion is used for all Interaction Losses.

**Total Loss.** The complete training objective combines all components through balanced weighting:

$$\mathcal{L}_{\mathrm{total}} = \mathcal{L}_{\mathrm{CRF}} + \lambda_{\mathrm{geo}}\mathcal{L}_{\mathrm{geo}} + \lambda_{\mathrm{inter}}\mathcal{L}_{\mathrm{inter}} \tag{10}$$

where the hyperparameters $\lambda_{\mathrm{geo}}$ and $\lambda_{\mathrm{inter}}$ are meticulously selected to regulate the magnitude of their corresponding terms.

## 4 RESULTS

**Datasets.** We train and evaluate DualFlow on three widely used two-person motion datasets spanning text-to-motion, music-to-dance, and multi-modal duet generation: **(1) InterHuman-AS** (Xu et al., 2024), an asymmetric extension of InterHuman (Liang et al., 2024) with actor-reactor labels, over 50K interaction clips across 11 action types (e.g., handshake, hug) and paired SMPL-X Pavlakos et al. (2019) sequences for modeling fine-grained interpersonal dynamics. **(2) DD100** (Siyao et al., 2024), featuring 100 duet dance routines (e.g., salsa, hip-hop, waltz) with high-resolution motion capture, paired music, and manually annotated dance structure for rhythm and style alignment. **(3) MDD** (Gupta et al., 2025), a large-scale multi-modal duet dance dataset with 10.3 hours of marker-based capture and 10K+ text annotations covering spatial relationships, choreography, movement quality, and music synchronization. Together, these datasets enable robust learning and evaluation of both interactive-reactive motion generation across multiple modalities.

**Implementation Details.** DualFlow consists of 20 cascaded blocks with 8 attention heads and dropout of 0.1. Both motion and conditioning inputs are projected into a 512-dimensional latent space, and each block's feedforward layer is set to size 1024. We use 4800-d Jukebox (Dhariwal et al., 2020) features for music and 768-d CLIP (ViT-L/14) (Radford et al., 2021) text embeddings. All cross-attention layers adopt Flash attention for faster processing. The stride values for the parallel convolution layers used are 7, 11 and 21. The model is trained with Contrastive Rectified Flow using 200 integration steps and a cosine $\beta$ scheduler. Training uses Adam with lr $2\times10^{-4}$, weight decay $2\times10^{-5}$, 1000 warm-up steps, batch size 32, for 5000 epochs. In the reactive setting, we use a 10-frame look-ahead. For classifier-free guidance, both modalities are masked 10% of the time, and text/music individually 20%. All hyperparameters were selected empirically on a held-out validation set.

**Evaluation Metrics.** We evaluate models using metrics adapted from text-to-motion (Liang et al., 2024) and music-to-motion (Siyao et al., 2024): *Frechet Inception Distance (FID)*: Distributional similarity between ground truth and generated motions; *Multimodal Distance (MM Dist)*: Text-motion alignment via feature distance; *R-Precision*: Text-motion alignment through retrieval accuracies within a batch; *Diversity*: Variety of generated motions regardless of conditions; *Multimodality (MModality)*: Diversity of generated motions under identical conditioning; *Beat Echo Degree (BED)*: Synchronization index of the both person's generated motion; *Beat-Alignment Score (BAS)*: Alignment between inflection points in motion and musical beats and Average Inference Time per Sentence (AITS) (Dai et al., 2024)

## 4.1 QUANTITATIVE METRICS

**Text & Music condition Motion Generation on MDD.** We evaluate DualFlow on MDD, InterHuman-AS, and DD100 using standard text-motion and music-motion metrics. As shown in Table 1, DualFlow consistently outperforms baselines across most metrics for duet and reactive tasks. In the interactive task, DualFlow (Both) achieves the highest R-Precision@3 (0.513) and lowest MMDist (0.513), indicating strong alignment with multimodal inputs. DualFlow (Text) records the best Beat-Align Score (BAS) at 0.215. While InterGen (Text) attains the best FID (0.405) and Diversity (1.405), DualFlow (Both) follows closely with an FID of 0.415 and a Diversity score of 1.307. For the reactive task, DualFlow (Both) leads in all R-Precision scores, FID (0.686), MMDist (1.056), and shows strong BAS (0.228). Although DuoLando (Both) has a slightly higher BED (0.395), DualFlow remains competitive at 0.215.

Table 1: Duet Generation results on MDD dataset with both text and music modalities. **Bold** for best, underline for second best.

| Methods | R-Precision↑ | | | FID↓ | MMDist↓ | Diversity→ | MModal↑ | BED↑ | BAS↑ |
|---|---|---|---|---|---|---|---|---|---|
| | **Top 1** | **Top 2** | **Top 3** | | | | | | |
| Ground Truth | 0.231 | 0.398 | 0.522 | 0.065 | 0.077 | 1.387 | - | 0.327 | 0.170 |
| **Duet Task** | | | | | | | | | |
| MDM(Text) | 0.082 | 0.124 | 0.192 | 1.420 | 2.133 | 1.216 | 0.811 | 0.211 | 0.186 |
| MDM(Music) | 0.041 | 0.102 | 0.135 | 2.241 | 2.471 | 1.192 | 0.411 | 0.210 | 0.192 |
| MDM(Both) | 0.061 | 0.108 | 0.163 | 1.739 | 2.244 | 1.235 | 0.787 | 0.194 | 0.190 |
| InterGen(Text) | 0.113 | 0.223 | 0.305 | **0.405** | 1.462 | 1.405 | 1.231 | **0.422** | 0.194 |
| InterGen(Music) | 0.023 | 0.067 | 0.088 | 2.014 | 2.526 | 1.300 | **1.768** | 0.364 | 0.163 |
| InterGen(Both) | 0.105 | 0.206 | 0.302 | 0.426 | 1.532 | 1.380 | 1.352 | 0.385 | 0.185 |
| DualFlow(Text) | **0.211** | 0.365 | 0.492 | 0.657 | 0.521 | 1.239 | 1.569 | 0.288 | **0.215** |
| DualFlow(Music) | 0.172 | 0.308 | 0.452 | 0.694 | 1.244 | 1.319 | 1.109 | 0.308 | 0.180 |
| DualFlow(Both) | 0.185 | **0.373** | **0.513** | 0.415 | **0.513** | **1.392** | 1.467 | 0.286 | 0.179 |
| **Reactive Task** | | | | | | | | | |
| DuoLando(Text) | 0.047 | 0.121 | 0.182 | 1.538 | 2.811 | 1.422 | - | 0.311 | 0.195 |
| DuoLando(Music) | 0.069 | 0.141 | 0.202 | 0.721 | 2.633 | **1.390** | - | 0.305 | 0.216 |
| DuoLando(Both) | 0.078 | 0.156 | 0.219 | 0.698 | 2.113 | 1.371 | - | **0.395** | 0.224 |
| DualFlow(Text) | 0.143 | 0.284 | 0.450 | 0.741 | 1.365 | 1.379 | 1.667 | 0.229 | **0.228** |
| DualFlow(Music) | 0.135 | 0.260 | 0.397 | 0.750 | 1.672 | 1.460 | **1.976** | 0.195 | 0.202 |
| DualFlow(Both) | **0.189** | **0.341** | **0.471** | **0.686** | **1.056** | 1.203 | 1.473 | 0.215 | 0.226 |

**Text-conditioned Motion Generation on InterHuman-AS.** Table 2 shows DualFlow significantly outperforms InterGen on R-Precision (Top-1: 0.437, Top-3: 0.681), with much lower MMDist (0.394) and the highest multimodality score (2.729). While InterGen has a slightly better FID (5.918 vs. 6.296), DualFlow offers better semantic and multimodal alignment. In the reactive task, we train our model with L=0 removing access to actor's intention (completely causal) defined as Unconstrained (UC) for fair comparison with ReGenNet(UC). DualFlow(UC) surpasses ReGenNet(UC) in R-Precision@3 (0.572 vs. 0.407), MMDist (6.314 vs. 6.860), Diversity (5.449 vs. 5.214) and Multimodality (2.502 vs. 2.391).

**Reactive Motion Generation on DD100.** Table 3 highlights DualFlow's performance across all metrics for reactive motion task. It achieves the best $FID_k$ (19.22), $FID_g$ (28.85), and $FID_{cd}$ (5.57), with strong diversity and rhythmic scores ($Div_k$: 11.01, BAS: 0.211). While Duolando leads in BED (0.285), DualFlow follows closely at 0.276, showing generative fidelity and collaborative modeling.

**Computational Complexity.** Figure 4 reports FID as a function of inference steps for DualFlow and InterGen. While InterGen requires more than 50 DDIM steps to reach high-quality performance, DualFlow achieves better FID with only 20 Rectified Flow (RF) steps. For a 10-second sequence at 30 FPS, the Average Inference Time per Sentence (AITS) on an RTX 5090 GPU is 1.92s for InterGen (50 DDIM steps) and 1.24s for DualFlow (20 RF steps), demonstrating improved efficiency under identical hardware and sequence length.

Table 2: Interactive Two-person Generation results conditioned on text modality for the InterHuman-AS dataset.

Figure 3: User study results

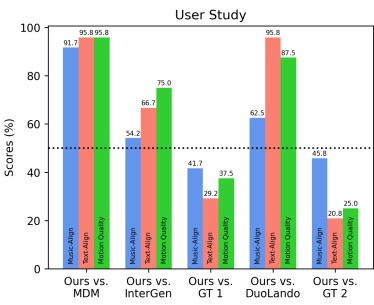

| Methods | R-Precision↑ | | | FID↓ | MMDist↓ | Diverse→ | MModal↑ |
|---------|------|------|------|------|---------|----------|---------|
| | Top 1 | Top 2 | Top 3 | | | | |
| Ground Truth | 0.452 | 0.610 | 0.701 | 0.273 | 3.755 | 7.948 | - |
| **Duet Task** | | | | | | | |
| InterGen | 0.371 | 0.515 | 0.624 | **5.918** | 5.108 | **7.387** | 2.141 |
| DualFlow | **0.437** | **0.558** | **0.681** | 6.296 | **4.394** | 7.116 | **2.729** |
| **Reactive Task** | | | | | | | |
| ReGenNet(UC) | - | - | 0.407 | **2.265** | 6.860 | 5.214 | 2.391 |
| DualFlow(UC) | **0.381** | **0.493** | **0.572** | 2.581 | **6.314** | 5.449 | **2.502** |
| DualFlow | 0.419 | 0.549 | 0.629 | 2.448 | 6.230 | 4.981 | 2.616 |

Table 3: Reactive Motion Generation results conditioned on text modality for the DD100 dataset.

Figure 4: FID vs. Steps

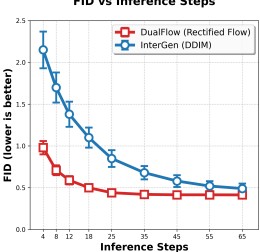

| Methods | Solo Metrics | | | | Interactive Metrics | | | Rhythmic |
|---------|-------|-------|-------|-------|--------|--------|--------|--------|
| | $FID_k↓$ | $FID_g↓$ | $Div_k↑$ | $Div_g↑$ | $FID_{cd}↓$ | $Div_{cd}↑$ | BED(↑) | BAS(↑) |
| Ground Truth | 6.56 | 6.37 | 11.31 | 7.61 | 3.41 | 12.35 | 0.5308 | 0.1839 |
| Bailando | 78.52 | 36.19 | **11.15** | 7.92 | 6643.31 | 52.50 | 0.1831 | 0.1930 |
| EDGE | 69.14 | 44.58 | 8.62 | 6.35 | 5894.45 | 60.62 | 0.1822 | 0.1875 |
| Duolando | 25.30 | 33.52 | 10.92 | **7.97** | 9.97 | 14.02 | **0.2858** | 0.2046 |
| DualFlow | **19.22** | **28.85** | 11.01 | 7.35 | **5.57** | 19.52 | 0.2767 | **0.2113** |

## 4.2 QUALITATIVE EVALUATION

Fig. 5 shows a Qualitative Comparison for two samples from MDD Dataset. While samples generated from both text and music condition-based InterGen and DualFlow models follow the text prompt, the motion quality of InterGen has reduced motion quality as circled, where the hands are flipping and the distance is increased. We also conduct a user study to qualitatively evaluate the motion sequences generated by our DualFlow framework in comparison with baseline methods on both tasks from the MDD dataset (details in Appendix). As shown in Fig.3, DualFlow outperforms the baseline methods across most comparisons, demonstrating superior alignment with both text and music, as well as high-quality motion generation.

## 4.3 ABLATION STUDY

We perform an ablation study on both the tasks (Table 4) to assess the impact of key DualFlow components. We compare the full model against four variants: (1) replacing Causal Look-Ahead (CLA) Attention with regular cross-attention (only for reactive setting), (2) removing RAG by replacing Retrieved Causal Attention with self-attention, (3) removing the triplet loss $\mathcal{L}_{triplet}$, and (4) substituting high-level Jukebox features with Mel-spectrograms. Results show clear performance drops across most metrics, highlighting the importance of anticipatory modeling, retrieval grounding, and rich audio features for high-quality reactive motion generation. Please refer to Appendix for more ablation results.

## 5 CONCLUSION

We introduced DualFlow, a unified rectified flow-based framework for efficient and expressive two-person 3D motion generation, supporting both interactive and reactive settings with text, music, and retrieved motion exemplars. Leveraging rectified flow enables faster sampling and lower latency than diffusion-based methods. Extensive evaluations on MDD, InterHuman-AS, and DD100 show superior performance in duet generation and reactive motion. DualFlow advances multi-modal two-person motion synthesis, opening new opportunities for immersive avatar interaction, intelligent

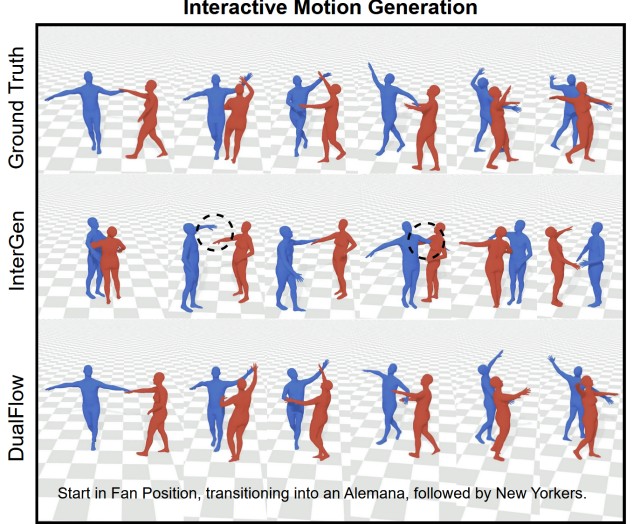

Figure 5: Comparing DualFlow with InterGen (interactive) and DuoLando (reactive) against ground truth on MDD Dataset. Black circles mark regions where baselines lose contact or produce distortions. InterGen shows artifacts like unnatural hand spacing, body interpenetration, and skipping the Alemana (follower's inside turn), while DuoLando shows incorrect leg initiation and head orientation. In contrast, DualFlow generates smooth, text-aligned choreography and coherent partner responses closely matching the ground truth. Supplementary video provides detailed visualizations.

Table 4: Ablation Study on MDD dataset (both text & music).

| Methods | R-Precision↑ | | | FID↓ | MMDist↓ | Diverse→ | MModal↑ | BED↑ | BAS↑ |
|---|---|---|---|---|---|---|---|---|---|
| | Top 1 | Top 2 | Top 3 | | | | | | |
| Ground Truth | 0.231 | 0.398 | 0.522 | 0.065 | 0.077 | 1.387 | - | 0.327 | 0.170 |
| **Interactive Task** | | | | | | | | | |
| DualFlow(w/o RAG) | 0.179 | 0.356 | 0.498 | 0.622 | 0.626 | 1.502 | 1.224 | 0.254 | 0.162 |
| DualFlow(w/o $\mathcal{L}_{\text{triplet}}$) | 0.158 | 0.297 | 0.412 | 0.783 | 0.818 | 1.433 | 0.844 | **0.291** | 0.169 |
| DualFlow(w/o $\mathcal{L}_{\text{sync}}$) | 0.182 | 0.369 | 0.509 | 0.472 | 0.590 | 1.224 | 1.340 | 0.277 | **0.182** |
| DualFlow(Spectral) | 0.172 | 0.321 | 0.477 | 0.647 | 0.633 | **1.383** | 1.114 | 0.255 | 0.158 |
| DualFlow(Jukebox) | **0.185** | **0.373** | **0.513** | **0.415** | **0.513** | 1.392 | **1.467** | 0.286 | 0.179 |
| **Reactive Task** | | | | | | | | | |
| DualFlow(w/o CLA) | 0.172 | 0.311 | 0.338 | 0.849 | **0.831** | 1.137 | 1.385 | 0.247 | 0.142 |
| DualFlow(w/o RAG) | **0.192** | **0.352** | **0.479** | 0.714 | 0.933 | 1.270 | 1.466 | 0.233 | 0.193 |
| DualFlow(w/o $\mathcal{L}_{\text{triplet}}$) | 0.153 | 0.292 | 0.308 | 0.885 | 1.328 | 1.664 | 1.007 | 0.204 | 0.186 |
| DualFlow(w/o $\mathcal{L}_{\text{sync}}$) | 0.166 | 0.311 | 0.453 | 0.774 | 1.112 | **1.429** | 1.233 | 0.235 | 0.202 |
| DualFlow(Spectral) | 0.162 | 0.301 | 0.468 | 0.721 | 0.965 | 1.261 | 1.401 | **0.255** | 0.162 |
| DualFlow(Jukebox) | 0.189 | 0.341 | 0.471 | **0.686** | 1.056 | 1.203 | **1.473** | 0.215 | **0.226** |

choreography, and responsive digital humans. Future work will explore improved interactive generation with newer flow-matching methods, real-time motion editing, and few-shot adaptation to novel styles and languages.

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

## A    LLM Disclosure

LLMs were only used to polish the text and proof read the paper for grammatical errors. They were not used to generate any metrics or citations.

## B    Reproducibility

Full code for this project along with the trained checkpoints for all tasks will be made open source and publicly available upon paper acceptance.

## C    LLM-based Decomposition

### C.1    Prompt Design

We design a structured prompting framework for the LLM, which is detailed as follows:

1. **System prompt:** We instruct the model with the following directive:
   *"As a professional dance movement analyst, please break down the given textual description of a duet dancing movement for {genre} into three focused descriptions: (1) Spatial Relationships: physical positioning, orientation, handhold (2) Body Movement: key gestures, actions, specific body part movements (3) Rhythm: tempo, timing, rhythmic dancing style and stepping. Please refer to the provided documents for guidance."*

2. **Few-shot Examples:** We provide a curated set of genre-specific examples (3 per genre) illustrating how input descriptions are manually decomposed into the three components. These examples were crafted by analyzing a diverse subset of textual annotations in the MDD dataset and annotating their corresponding focused descriptions through expert review.

3. **Reference Guidelines:** To promote interpretive consistency, we supply a supporting document containing structured definitions and keyword clusters describing typical language and semantic categories associated with each duet motion aspect.

### C.2    Generated Focused Descriptions

To enhance semantic grounding during retrieval, we leverage a Large Language Model (LLM) to decompose free-form textual prompts into structured, movement-relevant subcomponents. Drawing inspiration from Laban Movement Analysis (LMA), we extract three focused descriptions: *Spatial Relationship*, *Body Movement*, and *Rhythm*. This decomposition allows the system to perform more targeted motion retrieval by aligning each aspect of the prompt with corresponding motion features. By translating ambiguous or abstract user descriptions into focused representations, the objective for the LLM-based refinement is to improve both retrieval precision and downstream motion generation quality. Table 5 shows some of the examples for the focused textual descriptions for text prompts for the MDD Dataset.

### C.3    Validation of LLM-Based Semantic Decomposition

To verify that the LLM-generated spatial, body-movement, and rhythm descriptors accurately reflect the original human-written annotations, we randomly sampled 30 descriptions from MDD and InterHuman-AS and manually compared each decomposed attribute against the ground-truth text. Using our tuned GPT-4o prompt (Section C.1), two annotators independently evaluated consistency, correctness, and completeness, scoring each attribute on a 5-point scale (1 = incorrect, 5 = fully correct) based on consistency, correctness, and completeness. The decompositions showed high fidelity to the original descriptions, with accuracies of 96.1% for spatial relationships, 98.3% for body movement, and 86.9% for rhythm (overall 93.8%). We observed that in a few cases the LLM introduced rhythm-related terms were not explicitly present in the original text leading to lower validation accuracy. However, the use of music-derived features in our RAG module can help in naturally correcting such deviations by grounding rhythmic information. Overall, the results confirm that the

LLM reliably produces semantically aligned decompositions suitable for guiding retrieval in the RAG module.

Table 5: Examples of input text decomposed into three fine-grained, semantically focused descriptions using LLM for MDD Dataset.

| Text Description | Spatial Relationship | Body Movement | Rhythm |
|---|---|---|---|
| The leader switches the hand hold from left to right, leading the follower into a triple spin, maintaining a strong frame and connection. | The dancers are in an Open position with a Hand-to-hand connection. The leader switches the hand hold from left to right, maintaining a strong frame. They are facing each other during the transition. | The leader uses a strong frame to guide the follower into a triple spin. The follower's arms and torso are actively involved in the spinning motion, with medium energy. | The movement is executed at a fast tempo, with the triple spin occurring in quick succession, maintaining a continuous flow. |
| The dancers perform Jive Spanish Arms, maintaining a strong frame and connection, with the follower executing a controlled turn. | The dancers are in a Closed position, facing each other with a strong Hand-to-hand connection. The leader maintains a firm frame, guiding the follower through the movement. | The leader maintains a steady posture, using arms and shoulders to guide. The follower performs a controlled turn, involving a smooth rotation of the torso and arms, with medium energy. | The movement is executed at a fast tempo, characteristic of Jive, with a continuous and lively rhythm, ensuring the turn is seamlessly integrated into the dance sequence. |
| From a separated position, the leader draws the follower into a Closed Hand Hold, and they rotate clockwise together. | The dancers transition from a separated position to a Closed position with a Hand-to-hand connection. They are facing each other as they move into this position. | The leader initiates a drawing motion, pulling the follower towards him. Both dancers engage in a rotating movement, turning their bodies clockwise together. | The rotation is performed at a medium tempo, with a continuous and fluid motion as they move in sync with each other. |
| The leader brings the follower back with a circular motion, leading a head roll with his left hand, connecting it with a forward body roll for the follower. They then perform a basic step. | The dancers are in an Open position, with the leader facing the follower. They maintain a Hand-to-head connection as the leader guides the follower's head roll. | The leader uses his left hand to guide a head roll, involving the follower's head and neck. The follower transitions into a forward body roll, engaging the shoulders and torso. Both then perform a basic step, involving coordinated leg and foot movements. | The sequence begins with a medium-paced circular motion, transitioning into a fluid head and body roll. The basic step follows a steady, continuous tempo, maintaining rhythmic consistency. |
| The lead pulls the follow towards him, taking three steps, while the follow also takes three steps towards the lead. Both hands of both dancers are now connected. | The dancers are in a Closed position, facing each other. They have a Hand-to-hand connection with both hands engaged. | The lead and follow are both taking three steps towards each other. The movement involves the legs and feet, with a medium energy as they close the distance. | The steps are taken at a medium tempo, with each step evenly spaced, creating a continuous and synchronized rhythm between the dancers. |

# D  MODEL ARCHITECTURE DETAILS

The proposed framework for duet and reactive motion generation employs a rectified flow matching approach. Our model utilizes transformer-based architectures with multi-scale temporal modeling and attention mechanisms, supporting optional text and music conditioning. The following section discusses about specific modules used in detail.

## D.1  DUALFLOW BLOCK.

The DualFlow block applies multi-scale temporal convolutions with learnable gating:

$$\mathbf{f}_b^{(k)} = \text{GELU}(\text{Conv1D}_k(\mathbf{z}_b^{(j)\top}))^\top, \quad k \in \{1, 2, 3\}, \quad \mathbf{z}_b^{(j')} = \mathbf{z}_b^{(j)} + \sum_{k=1}^{3} \gamma_k \mathbf{f}_b^{(k)},$$

Each block applies a sequence of self- and cross-attention layers with residual connections and LayerNorm conditioning using the text latent $\mathbf{z}_d$. Let $\text{LN}(\cdot, \mathbf{z}_d)$ denote LayerNorm with text-conditioned shift/scale, and $\text{Attn}(\mathbf{Q}, \mathbf{K}, \mathbf{V}) = \text{softmax}\left(\frac{\mathbf{Q}\mathbf{K}^\top}{\sqrt{d}}\right)\mathbf{V}$. The transformations applied are

Self-Attention (equation 11), Music Cross Attention (equation 12), Motion Cross Attention (equation 13), Retrieval Cross Attention (equation 14), and Feedforward (FFN) Layer (equation 15):

$$\mathbf{z}_a^{(j,1)} = \mathbf{z}_a^{(j')} + \text{Attn}\big(\mathbf{Q} = W_Q^{\text{sa}}\,\text{LN}(\mathbf{z}_a^{(j')}, \mathbf{z}_d), \mathbf{K} = W_K^{\text{sa}}\,\text{LN}(\mathbf{z}_a^{(j')}, \mathbf{z}_d), \mathbf{V} = W_V^{\text{sa}}\,\text{LN}(\mathbf{z}_a^{(j')}, \mathbf{z}_d)\big) \tag{11}$$

$$\mathbf{z}_a^{(j,2)} = \mathbf{z}_a^{(j,1)} + \text{Attn}\big(\mathbf{Q} = W_Q^{m_1}\,\text{LN}(\mathbf{z}_a^{(j,1)}, \mathbf{z}_d), \mathbf{K} = W_K^{m_1}\,\mathbf{z}_m, \ \mathbf{V} = W_V^{m_1}\,\mathbf{z}_m\big) \tag{12}$$

$$\mathbf{z}_a^{(j,3)} = \mathbf{z}_a^{(j,2)} + \text{Attn}\big(\mathbf{Q} = W_Q^{m_2}\,\text{LN}(\mathbf{z}_a^{(j,2)}, \mathbf{z}_d), \mathbf{K} = W_K^{m_2}\,\mathbf{z}_b^{(j,2)}, \ \mathbf{V} = W_V^{m_2}\,\mathbf{z}_b^{(j,2)}\big) \tag{13}$$

$$\mathbf{z}_a^{(j,4)} = \mathbf{z}_a^{(j,3)} + \text{Attn}\big(\mathbf{Q} = W_Q^R\,\text{LN}(\mathbf{z}_a^{(j,3)}, \mathbf{z}_d), \mathbf{K} = W_K^R\,\mathbf{z}_R, \ \mathbf{V} = W_V^R\,\mathbf{z}_R\big) \tag{14}$$

$$\mathbf{z}_a^{(j+1)} = \mathbf{z}_a^{(j,4)} + \text{FFN}\big(\text{LN}(\mathbf{z}_a^{(j,4)}, \mathbf{z}_d)\big). \tag{15}$$

with symmetric updates for $\mathbf{z}_b^{(j)}$.

## D.2 INTERACTIVE SETTING

The flow dynamics are defined as:
$$\mathbf{x}(t) = [\mathbf{x}_a(t); \mathbf{x}_b(t)], \quad \mathbf{v}_\theta(\mathbf{x}(t), t, c) = [\mathbf{v}_{\theta,a}(\mathbf{x}(t), t, c); \mathbf{v}_{\theta,b}(\mathbf{x}(t), t, c)].$$

The final motion latents $\mathbf{z}_a^{(N)}$ and $\mathbf{z}_b^{(N)}$ are mapped to velocity fields
$$\mathbf{v}_{\theta,a} = \text{Linear}(\mathbf{z}_a^{(N)}), \quad \mathbf{v}_{\theta,b} = \text{Linear}(\mathbf{z}_b^{(N)}), \tag{16}$$

concatenated as
$$\mathbf{v}_\theta = [\mathbf{v}_{\theta,a}; \mathbf{v}_{\theta,b}] \in \mathbb{R}^{B \times T \times 524}. \tag{17}$$

## D.3 REACTIVE SETTING

For reactive motion generation, our model generates the reactor's motion $\mathbf{x}_b$ conditioned on the actor's fixed motion $\mathbf{x}_a$, with the flow dynamics defined as:
$$\mathbf{x}(t) = [\mathbf{x}_a; \mathbf{x}_b(t)], \quad \mathbf{v}_\theta(\mathbf{x}(t), t, c) = [\mathbf{0}; \mathbf{v}_{\theta,\text{reactor}}(\mathbf{x}(t), t, c)].$$

The Motion Cross Attention gets replaced by Causal Cross Attention in the DualFlow block for this setting. The final reactor latent $\mathbf{z}_b^{(N)}$ is mapped to the velocity field $\mathbf{v}_{\theta,\text{reactor}} = \text{Linear}_L^{262}(\mathbf{z}_b^{(N)})$, and the output is $\mathbf{v}_\theta = [\mathbf{0}; \mathbf{v}_{\theta,\text{reactor}}] \in \mathbb{R}^{B \times T \times 524}$. During inference, the initial state is $\mathbf{x}(0) = [\mathbf{x}_a; \mathbf{z}_b]$, where $\mathbf{z}_b \sim \mathcal{N}(\mathbf{0}, \mathbf{I})$.

## D.4 CAUSAL CROSS ATTENTION WITH LOOK-AHEAD

The Causal Cross Attention module enables the reactor to condition on the actor's motion while preserving temporal causality and allowing limited future anticipation. For reactor motion latent $\mathbf{z}_b^{(j,2)}$ and fixed actor motion latent $\mathbf{z}_a$ from DualFlow block $j$, we construct query, key, and value matrices as $\mathbf{Q} = \mathbf{z}_b^{(j,2)}\mathbf{W}_Q$, $\mathbf{K} = \mathbf{z}_a\mathbf{W}_K$, and $\mathbf{V} = \mathbf{z}_a\mathbf{W}_V$, where $\mathbf{W}_Q, \mathbf{W}_K$, and $\mathbf{W}_V \in \mathbb{R}^{L \times d_k}$ are learned projection matrices. The causal mask with look-ahead parameter $L$ uses an upper triangular mask such that reactor's motion attends to past and only $L$ future frames of the actor's motion, implemented as $\mathbf{M}_{i,j} = 1$ if $j \leq i + L$ and $\mathbf{M}_{i,j} = 0$ otherwise. The attention computation follows:

$$\text{CausalCrossAttention}(\mathbf{Q}, \mathbf{K}, \mathbf{V}) = \text{softmax}\left(\frac{\mathbf{Q}\mathbf{K}^T}{\sqrt{d_k}} \odot \mathbf{M} + (1 - \mathbf{M}) \cdot (-\infty)\right) \mathbf{V}$$

where $\odot$ denotes element-wise multiplication. This formulation ensures temporally aligned and context-aware reactive generation, enabling natural reactive responses that align with the actor's intended trajectory without violating temporal consistency.

## D.5 MODEL PARAMETERS

**Loss Weighting Values** We assign higher weights to geometric losses for velocity ($\lambda_{\text{vel}} = 30$) and foot contact ($\lambda_{\text{foot}} = 30$), moderate weight for bone length consistency ($\lambda_{\text{BL}} = 10$), and emphasize inter-dancer synchronization ($\lambda_{\text{sync}} = 5$). Affinity and distance are equally weighted ($\lambda_{\text{DM}} = 3$), while orientation receives a minimal weight ($\lambda_{\text{RO}} = 0.01$). These settings ensure anatomically plausible, temporally smooth, and well-coordinated duet motions.

# E QUANTITATIVE EVALUATION

We further conduct ablations to study model design choices in Table. 6: (1) replacing the three temporally scaled parallel convolutions with a single convolution, (2) reducing the number of transformer blocks to 10 and 15 (from 20), (3) lowering the latent dimension to 128 and 256 (from 1024) and (4) changing the Look-Ahead parameter L to 0 and 20. These variants consistently show performance drops across most metrics, highlighting the benefit of the full architecture. Performance decrease in different settings shows the importance of 3 parallel temporal Convs, using 20 blocks, 515 Latent dimension and Look-Ahead parameter L = 10 frames. Here, **Bold** indicates the best result and Underline indicates the second best result.

Table 6: Ablation study results for Reactive Setting on the MDD dataset

| Methods | R-Precision↑ | | | FID↓ | MMDist↓ | Diversity→ | MModal↑ | BED ↑ | BAS↑ |
|---|---|---|---|---|---|---|---|---|---|
| | Top 1 | Top 2 | Top 3 | | | | | | |
| Ground Truth | 0.231 | 0.398 | 0.522 | 0.065 | 0.077 | 1.387 | – | 0.327 | 0.170 |
| DualFlow (one conv) | 0.172 | 0.311 | 0.338 | 0.595 | 0.582 | 1.288 | 1.385 | 0.266 | 0.142 |
| DualFlow (10 blocks) | 0.160 | 0.313 | 0.452 | 0.683 | 0.654 | 1.215 | 1.222 | 0.259 | 0.159 |
| DualFlow (15 blocks) | 0.175 | 0.357 | **0.521** | 0.482 | 0.627 | 1.211 | 1.402 | 0.270 | 0.163 |
| DualFlow (128 latent) | 0.108 | 0.284 | 0.414 | 0.966 | 0.834 | 1.277 | 1.091 | 0.273 | 0.141 |
| DualFlow (256 latent) | 0.168 | 0.342 | 0.468 | 0.642 | 0.681 | 1.245 | 1.328 | **0.291** | 0.163 |
| DualFlow (L=0) | 0.162 | 0.322 | 0.455 | 0.574 | 0.663 | 1.292 | 1.274 | 0.241 | 0.152 |
| DualFlow (L=20) | 0.181 | 0.366 | 0.507 | 0.497 | 0.542 | **1.322** | 1.393 | 0.258 | 0.167 |
| DualFlow | **0.185** | **0.373** | 0.513 | **0.415** | **0.513** | 1.307 | **1.467** | 0.286 | **0.179** |

**Ablation for RAG.** We also perform ablations to critically evaluate the role of retrieval-augmented components across both the settings in driving DualFlow's performance in Table. 7. For the cases where different retrieval components are ablated, value of k is set to be 5. For no text-decompose setting of RAG, we directly perform retrieval on original text descriptions and music features in order to understand the benefit from text decomposition.

In the interactive setting, removing any individual retrieval cue consistently degrades semantic alignment and motion quality, with the largest drops observed when all retrieval components are removed. Increasing the number of retrieved samples shows a clear sweet spot where k = 5 achieves the best R-Precision, FID, and Multi-modality scores, indicating that moderately diverse retrieved context helps the model ground its generation without introducing noise. Interestingly, k = 3 already provides a substantial boost over no retrieval, but larger retrieval depth (k = 7) offers diminishing returns and slightly worse fidelity, suggesting an over-saturation of context. Using no textual decomposition setting provides similar results as removing Music-based retrieval but having retrieval on decomposed text components.

In contrast, the reactive setting exhibits a different trend. Because the follower must respond tightly to the leader's motion in real time, excessive retrieval diversity can introduce temporal drift. It can be seen that k = 3 provides the strongest semantic alignment, outperforming both lower (k = 1) and higher (k = 5, 7) retrieval depths. Additionally, removing music-based retrieval surprisingly improves R-Precision and MM-Distance, suggesting that in tightly synchronized partner interactions, leader motion cues dominate over rhythmic cues for determining the follower's behavior. Using no textual decomposition RAG setting performs better than text-retrieval ablated version but performs more comparable to text rhythm component ablated version.

Table 7: Ablation Study on RAG in DualFlow on the MDD dataset

| Methods | R-Precision↑ | | | FID↓ | MMDist↓ | Diverse→ | MModal↑ | BED↑ | BAS↑ |
|---|---|---|---|---|---|---|---|---|---|
| | Top 1 | Top 2 | Top 3 | | | | | | |
| Ground Truth | 0.231 | 0.398 | 0.522 | 0.065 | 0.077 | 1.387 | - | 0.327 | 0.170 |
| **Interactive Task** | | | | | | | | | |
| w/o RAG ($R_i^S, R_i^B, R_i^R, R_i^M$) | 0.179 | 0.356 | 0.498 | 0.622 | 0.626 | 1.502 | 1.224 | 0.254 | 0.162 |
| w/o Text-based Retrieval ($R_i^S, R_i^B, R_i^R$) | 0.181 | 0.361 | 0.503 | 0.541 | 0.574 | 1.441 | 1.351 | 0.263 | 0.171 |
| w/o $R_i^S$ | 0.180 | 0.359 | 0.501 | 0.529 | 0.566 | 1.431 | 1.432 | 0.289 | 0.169 |
| w/o $R_i^B$ | 0.182 | 0.364 | 0.506 | 0.520 | 0.559 | 1.422 | 1.419 | 0.272 | 0.172 |
| w/o $R_i^R$ | 0.181 | 0.362 | 0.504 | 0.512 | 0.553 | 1.416 | 1.441 | 0.267 | 0.177 |
| w/o Music-based Retrieval ($R_i^M$) | 0.183 | 0.368 | 0.509 | 0.498 | 0.541 | 1.406 | 1.452 | 0.268 | 0.164 |
| w RAG (no text-decompose) | 0.183 | 0.352 | 0.501 | 0.508 | 0.552 | 1.409 | 1.444 | 0.287 | 0.178 |
| w RAG (k=1) | 0.181 | 0.360 | 0.503 | 0.449 | 0.535 | 1.381 | 1.437 | 0.279 | 0.176 |
| w RAG (k=3) | 0.184 | 0.372 | 0.512 | 0.418 | 0.521 | **1.386** | 1.452 | **0.291** | 0.178 |
| w RAG (k=5) | **0.185** | **0.373** | **0.513** | **0.415** | **0.513** | 1.392 | **1.467** | 0.286 | **0.179** |
| w RAG (k=7) | 0.183 | 0.369 | 0.509 | 0.438 | 0.527 | 1.407 | 1.445 | 0.282 | 0.177 |
| **Reactive Task** | | | | | | | | | |
| w/o RAG ($R_i^S, R_i^B, R_i^R, R_i^M$) | 0.192 | 0.352 | 0.479 | 0.714 | 0.933 | **1.270** | 1.466 | 0.233 | 0.193 |
| w/o Text-based Retrieval ($R_i^S, R_i^B, R_i^R$) | 0.181 | 0.334 | 0.451 | 0.752 | 0.984 | 1.196 | 1.312 | 0.221 | 0.217 |
| w/o $R_i^S$ | 0.182 | 0.321 | 0.449 | 0.703 | 0.956 | 1.243 | 1.429 | **0.246** | 0.224 |
| w/o $R_i^B$ | 0.182 | 0.322 | 0.451 | 0.699 | 0.948 | 1.255 | 1.442 | 0.239 | 0.198 |
| w/o $R_i^R$ | 0.186 | 0.334 | 0.468 | 0.697 | 0.932 | 1.249 | 1.451 | 0.231 | 0.208 |
| w/o Music-based Retrieval ($R_i^M$) | **0.194** | **0.369** | **0.492** | 0.692 | 0.921 | 1.238 | 1.438 | 0.228 | 0.189 |
| w RAG (no text-decompose) | 0.185 | 0.336 | 0.473 | 0.696 | 0.933 | 1.252 | 1.442 | 0.221 | 0.208 |
| w RAG (k=1) | 0.190 | 0.348 | 0.457 | 0.707 | 0.978 | 1.223 | 1.469 | 0.221 | 0.209 |
| w RAG (k=3) | 0.193 | 0.367 | 0.483 | 0.693 | 0.962 | 1.217 | 1.471 | 0.224 | 0.212 |
| w RAG (k=5) | 0.189 | 0.341 | 0.471 | **0.686** | 1.056 | 1.203 | **1.473** | 0.215 | **0.226** |
| w RAG (k=7) | 0.188 | 0.336 | 0.459 | 0.699 | 0.989 | 1.229 | 1.470 | 0.218 | 0.223 |

Table 8: Ablation Study on Synchronization Loss on the MDD dataset.

| Methods | R-Precision↑ | | | FID↓ | MMDist↓ | Diverse→ | MModal↑ | BED↑ | BAS↑ |
|---|---|---|---|---|---|---|---|---|---|
| | Top 1 | Top 2 | Top 3 | | | | | | |
| Ground Truth | 0.231 | 0.398 | 0.522 | 0.065 | 0.077 | 1.387 | - | 0.327 | 0.170 |
| **Interactive Task** | | | | | | | | | |
| DualFlow(w/o $\mathcal{L}_{sync}$) | 0.182 | 0.369 | 0.509 | 0.472 | 0.590 | 1.224 | 1.340 | 0.277 | **0.182** |
| DualFlow(w $\mathcal{L}_{sync}$ w/o $w_d$) | 0.181 | 0.365 | 0.502 | 0.465 | 0.592 | 1.318 | 1.322 | 0.268 | 0.163 |
| DualFlow(w $\mathcal{L}_{sync}$ w/o $w_j$) | 0.184 | 0.372 | 0.511 | 0.432 | 0.538 | **1.385** | 1.435 | **0.292** | 0.180 |
| DualFlow (w $\mathcal{L}_{sync}$) | **0.185** | **0.373** | **0.513** | **0.415** | **0.513** | 1.392 | **1.467** | 0.286 | 0.179 |
| **Reactive Task** | | | | | | | | | |
| DualFlow(w/o $\mathcal{L}_{sync}$) | 0.166 | 0.311 | 0.453 | 0.774 | 1.112 | 1.429 | 1.233 | **0.235** | 0.202 |
| DualFlow(w $\mathcal{L}_{sync}$ w/o $w_d$) | 0.168 | 0.314 | 0.459 | 0.763 | 1.101 | **1.381** | 1.260 | 0.231 | 0.194 |
| DualFlow(w $\mathcal{L}_{sync}$ w/o $w_j$) | 0.181 | 0.334 | 0.467 | 0.712 | 1.064 | 1.312 | 1.431 | 0.212 | 0.214 |
| DualFlow | **0.189** | **0.341** | **0.471** | **0.686** | **1.056** | 1.203 | **1.473** | 0.215 | **0.226** |

**Ablation on Synchronization Loss.** Table. 8 shows further ablation analysis on the proposed Synchronization Loss. It can be seen that having $\mathcal{L}$sync plays a crucial role in improving both semantic alignment and inter-person coordination for duet motion generation. Removing the loss entirely leads to clear degradation across all metrics in both interactive and reactive settings, with notably higher FID & MMDist and reduced R-Precision. The distance weighting term $w_d$ and the anatomical weighting term $w_j$ contribute complementary benefits. Omitting $w_d$ harms spatial coherence and leads to greater overall performance degradation, whereas omitting $w_j$ primarily reduces semantic consistency and relational fidelity reflected in lower BED, BAS, and MModal, and thus performs slightly worse than the complete version. The full formulation consistently achieves the strongest performance, yielding the best balance of retrieval alignment (R-Precision), motion realism (FID), Diversity, Multimodality, and inter-person synchronization. These results validate that both weighting components are necessary and that $\mathcal{L}$sync meaningfully strengthens DualFlow's ability to model coordinated two-person motion.

**Model Parameters Comparison.** The adapted InterGen model—augmented with an additional music-attention layer to support both motion and music conditioning—contains 224M trainable parameters. InterGen's architecture packs two sub-blocks (each comprising two attention layers) into a single block, yielding a total of 8 blocks, i.e., $8 \times 2$ sub-blocks $\times 3$ attention layers per sub-

block (after adding music attention), resulting in 48 attention layers overall. In contrast, DualFlow employs 20 blocks, each containing four attention layers, amounting to 80 attention layers and a total of 456M trainable parameters. The increased capacity in DualFlow primarily arises from the added retrieval-augmented generation (RAG) module, which introduces additional attention layers and projection components necessary for multi-modal retrieval integration.

## F    QUALITATIVE EVALUATION

**User Study Details.** A total of 24 participants were recruited for the study. Each participant is shown 15 pairs of rendered videos (3 per experiment), with each video lasting less than 10 seconds. Each pair consists of one motion sequence generated by DualFlow and the other by either a baseline method or the ground truth (when available). To ensure unbiased evaluation, the order of videos within each pair is randomized, and no method labels are revealed. For each video pair, participants are asked to answer three key questions: *(1) Which motion better aligns semantically with the textual description? (2) Which motion is better synchronized with the musical beats? (3) Which motion has higher overall quality (e.g., naturalness, smoothness etc)?* Fig.6 shows the User Study Form we used.

Fig. 6 illustrates the User Study Form presented to participants during the human evaluation study. Clear and detailed guidelines were provided at the beginning of the form, explaining the evaluation criteria. Participants were then asked to watch two videos: one containing motion from either a Baseline model or the Ground Truth, and the other generated using our DualFlow model. The identity of each video (i.e., whether it was from the DualFlow model or the comparison method) was not disclosed to the participants. For each experimental condition, participants viewed and evaluated three distinct pairs of videos.

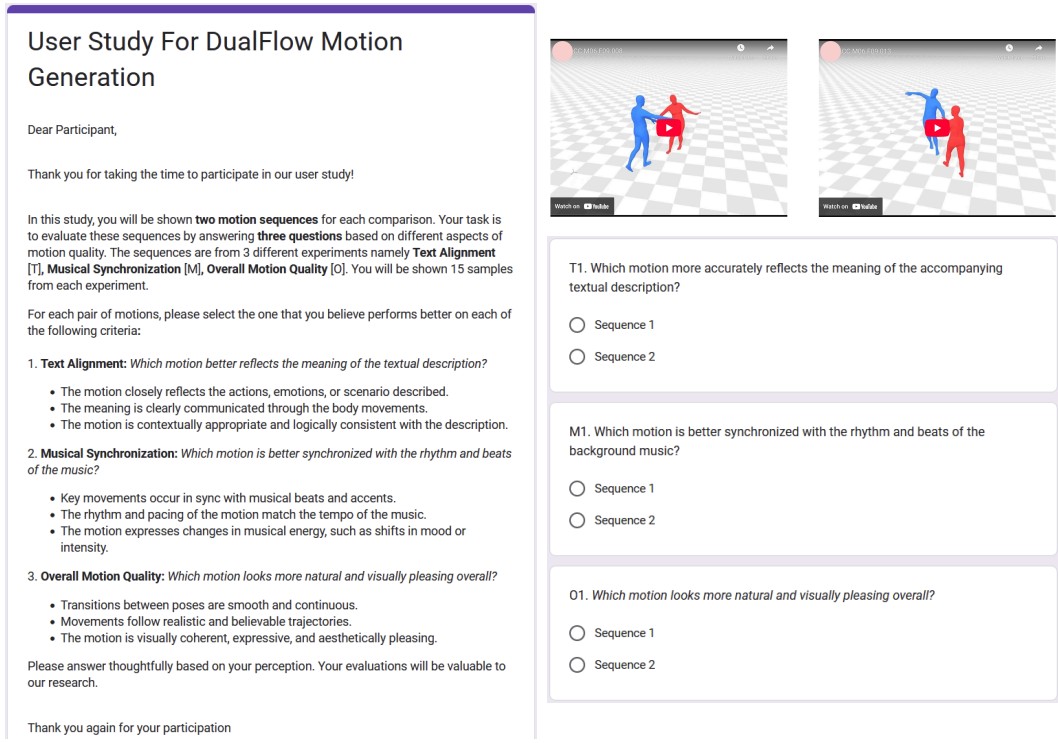

Figure 6: User Study Google Form

## G    LIMITATIONS AND FUTURE WORK

In this section, we discuss the limitations of the DualFlow model along with several observed failure cases followed by potential avenues for improvement. (1) The effectiveness of RAG-based mo-

tion alignment is dependent on the quality and relevance of the retrieved samples. In cases where the input text, leader motion, or music cues are ambiguous or underspecified, the RAG module may retrieve semantically mismatched neighbors. This semantic retrieval misalignment can cause stylistic drift or generate motions that deviate from the intended interaction attributes, particularly for prompts involving abstract descriptions or uncommon dance style/movement. (2) In the reactive setting, DualFlow occasionally struggles to maintain precise physical coordination between partners. We observe minor hand–hand or torso–torso penetrations during close-contact sequences or under rapid leader movements, likely due to the absence of explicit modeling of contact-based physical constraints. (3) Since retrieval operates over short, localized motion segments, directly generating long sequences can accumulate temporal drift, leading to weakened structural consistency or off-beat rhythmic alignment over extended durations.

The above limitations point to several promising directions for future work. Improving retrieval quality through learned semantic re-ranking, cross-modal retrieval scoring, or uncertainty-aware retrieval could reduce misalignment and make the system more robust to ambiguous input cues. Incorporating contact-based physical constraints as a loss function may help enforce more accurate hand and body coordination in close-contact motions. Finally, addressing long-term drift may benefit from introducing hierarchical temporal modeling, where high-level rhythmic or structural constraints guide long-range consistency, while DualFlow refines short-term details. Broadening the retrieval corpus to incorporate more diverse styles and partner interaction patterns may further enhance robustness. Together, these directions offer a path toward more physically grounded, semantically aligned and temporally coherent two-person motion generation.

