# OpenReview forum: "Unified Multi-Modal Interactive and Reactive 3D Motion Generation via Rectified Flow"
_ICLR.cc/2026/Conference — ICLR 2026 Poster_

### Official Review · Reviewer_9pa5 · 2025-10-26

**Soundness:** 3
**Presentation:** 3
**Contribution:** 3
**Rating:** 4
**Confidence:** 3

**Summary:**

This paper presents a unified framework for two-person motion generation from texts or/and music. DualFlow employs a retrieval-augmented generation module for semantic alignment. It also uses a contrastive rectified flow to achieve good sampling quality. Experiments on three datasets, InterHuman-AS, DD100, and MDD all shows DualFlow outperforms most baseline methods, especially in terms of the semantic alignment, showing the effectiveness of the proposed RAG, and contrastive learning object.

**Strengths:**

+ A new SOTA performance is achieved with the proposed method, with cutting-edge techniques, including RAG, contrastive rectified flow. It will serve as a better and stronger baseline in the field of two-person motion generation.
+ Experiments are extensive. Three benchmarks and ablation studies all shows the effectiveness of the proposed method.

**Weaknesses:**

- The paper looks very much like an "updated and modern version" of motion diffusion model, with some fancy new techniques from broader diffusion community to boost the performance, but is short of insights in motion generation itself, thus makes me feel lack of novelty.
- I did not see many insights in the "two-person" motion generation. Joint modeling of two-person motions and the synchronization loss are pretty standard in two-person motion generation. RAG and contrastive rectified flow are very general method and can be applied on single-person motion generation.

Typo:
- Line 295: L_{sync} => L_{inter}

**Questions:**

N/A

---

> ### Author Response · Authors · 2025-11-27
> **Response to Reviewer 9pa5 (Part 1/3)**
>
> We thank the reviewer for the positive assessment of presentation and contribution of our work. We address the reviewer’s concerns on novelty below.
>
> **(W1)** We understand the concern and would like to clarify that DualFlow is not an incremental update of MDM. While we build upon generative modeling literature, the architectural, methodological, and task-level innovations are specific to two-person, multi-modal motion generation, a domain that poses unique challenges far beyond those addressed in single-person diffusion models.
>
> Specifically:
> 1. **Unified modeling of interactive and reactive two-person motion.**
>  Prior work treats these as two disjoint problems, requiring separate architectures, training pipelines, and conditioning strategies. DualFlow is the first to unify both, enabled by a masking mechanism and task-adaptive switchable attention module. While both branches operate symmetrically in interactive setting; the actor branch is masked in reactive setting and the reactor branch employs a Causal Cross Attention module with Look-Ahead L, replacing Motion Cross Attention, to condition on the actor’s motion. This architectural design is novel specific to two-person generation and does not exist in single-person diffusion or flow-based models.
> 2. **First interaction-aware Retrieval-Augmented Generation (RAG) for two-person motion.**
>  Existing RAG methods (ReMoDiffuse[6], MoRAG [7] etc.) apply retrieval only to single-person motion, where no relational or inter-person grounding is required. Extending RAG to two-person settings is non-trivial because the retrieval must capture spatial relationships between agents, coordinated body movement patterns, rhythmic cues in music, and interaction semantics. We achieve this by retrieving motion samples based on LLM-derived semantic decompositions and music features. These samples are injected through retrieval-based cross-attention within each DualFlow block, providing explicit grounding for interpersonal structure and rhythm, which traditional embeddings struggle to represent.
> We have now added a new ablation study (Table 7, Section E of the Supplementary) evaluating each retrieval component independently and observed new insights:
> - **Interactive setting:** Removing any retrieval component consistently reduces semantic alignment, fidelity, and multi-modal consistency.
> - **Reactive setting:** Model without music-based retrieval perform best, but removing text-based retrieval still degrades performance.
> 3. **Contrastive Rectified Flow for high-dimensional two-person motion.**
> Two-person motion has substantially higher dimensionality and relational complexity than single-person motion. Introducing a contrastive (triplet) objective within rectified flow directly addresses mode collapse and weak semantic alignment in multi-agent contexts. This component is not present in prior diffusion/flow literature on motion generation.
> 4. **Empirical insights through extensive evaluations.**
> Our extensive Ablations (Table 5, 6, 7) of different components provide key insights of benefit from each component of the architecture. Our new RAG ablations reveal previously unreported behavior that music retrieval benefits interactive generation but not reactive generation providing new observation about the semantics of two-person coordination.

---

> > ### Author Response · Authors · 2025-11-27
> > **Response to Reviewer 9pa5 (Part 2/3)**
> >
> > **(W2)** We thank the reviewer for raising this concern and would like to clarify the specific insights and non-trivial extensions that DualFlow introduces for two-person motion generation. Although some high-level components (e.g., RAG, contrastive flow) exist in broader generative modeling, their adaptation to two-person generation with asymmetric roles, multimodal cues, and mutual dependencies is non-trivial. Our contributions are designed explicitly to address challenges unique to two-person motion:
> > 1. **Unified modeling of interactive and reactive generation is not standard in two-person motion.** Existing two-person approaches are strictly based on one of the tasks: Interactive-only (InterGen[1], DuetGen[2]) or Reactive-only (ReGenNet[3], DuoLando[4]) These methods require distinct architectures, conditioning signals and training pipelines. DualFlow is the first to provide a single architecture that can switch dynamically between interactive (bidirectional coordination-based generation) and reactive (asymmetric actor-reactor generation). The masking mechanism and task-adaptive switchable attention module are designed specifically to make both tasks compatible within a single model.
> > 2. **Interaction-aware RAG is fundamentally different from single-person RAG.** We agree that RAG is a general technique, but applying it to two-person motion requires  modelling interpersonal spatial relationships, coordinated body-movements, role-specific actions, rhythmic alignment, relative positioning and handholds. None of these factors exist in single-person motion. To address this, DualFlow introduces:
> > - LLM-based semantic decomposition (Spatial / Body / Rhythm) that isolates interaction-specific attributes.
> > - Retrieval-based cross-attention which semantically grounds generation based on interaction-specific attributes & music features.
> > To the best of our knowledge, this is the first RAG framework designed for two-person generation, and cannot be directly obtained from single-person generation RAG frameworks.
> > 3. **Contrastive Rectified Flow enforces semantic consistency in two person motion.**
> > Our contrastive rectified flow framework enforces semantic consistency through mechanisms tailored to two-person motion: We construct triplets that preserve actor-reactor relationships, ensuring the model learns how the actor initiates and the reactor responds. The contrastive loss enforces that predicted velocities for both persons remain semantically consistent with their interaction context. Our sampling strategy for negatives specifically targets violations of spatial relationships between the two agents. By contrasting semantically matched versus mismatched pairs, we stabilize the coupled flow in the joint motion space, preventing the two agents from diverging. This approach is specific to high-dimensional coupled motion spaces and is not present in prior flow or diffusion works. We have provided further clarification on how velocity samples are selected in section 3.4
> >  4. **Synchronization loss is an added novel component to the commonly used interaction loss**
> > While interaction-based losses have appeared in prior two-person motion works, we clarify that none of these methods (e.g., InterGen, ReGenNet, DuoLando) introduce an explicit synchronization loss that models weighted, joint-to-joint temporal coordination across two bodies. Existing approaches typically rely on heuristic proximity thresholds or coarse interaction penalties, which do not encode fine-grained relational structure or differentiate the anatomical importance of different joint pairs. In contrast, our synchronization loss is explicitly formulated to enforce relative-motion consistency between semantically linked joints by combining spatial distance weighting and anatomy-aware relevance weighting:
> > \begin{equation}
> > L_{\text{sync}} = \sum{} w_{\text{d}}(j_1,j_2), w_{\text{j}}(j_1,j_2), \left|d_{\text{p}}(j_1,j_2) - d_{\text{gt}}(j_1,j_2)\right|^2 ,
> > \end{equation}
> > where $d_{\text{p}}(j_1,j_2)$ and $d_{\text{gt}}(j_1,j_2)$ denote predicted and ground-truth inter-person joint distances.
> > We have added more details in Section 3.5 about how these weight terms are selected described as follows:
> > The distance-based weight $w_{\text{d}}$ emphasizes joint pairs that naturally maintain close spatial relationships:
> > \begin{equation}
> > w_{\mathrm{d}}(j_1, j_2)
> > = e^{\left(-\alpha , | d_{\mathrm{gt}}(j_1, j_2) |\right)}
> > \end{equation}
> > while the joint-type weight $w_{\text{j}}$ incorporates anatomical relevance for different joint groups. Complete details are added in Section 3.5 under "Interaction Losses".
> >
> > This formulation encourages high-frequency synchrony in key interaction regions (e.g., hands, upper body) while maintaining global relational structure between the two bodies (existing interaction losses do not explicitly provide)

---

> > > ### Author Response · Authors · 2025-11-27
> > > **Response to Reviewer 9pa5 (Part 3/3)**
> > >
> > > **(W3)** We have now corrected this typographical error in the paper.
> > >
> > > Overall, the novelty of DualFlow lies not in applying existing diffusion improvements, but in new architectural, retrieval, and training insights specifically tailored to two-person, multi-modal, interactive/reactive motion synthesis. We hope these clarifications help the reviewer better appreciate the originality of the work.
> > >
> > > **References:**
> > >
> > > [1] Liang H, Zhang W, Li W, et al. Intergen: Diffusion-based multi-human motion generation under complex interactions[J]. International Journal of Computer Vision, 2024, 132(9): 3463-3483.
> > >
> > > [2] Anindita Ghosh, Bing Zhou, Rishabh Dabral, Jian Wang, Vladislav Golyanik, Christian Theobalt, Philipp Slusallek, and Chuan Guo. Duetgen: Music driven two-person dance generation via hierarchical masked modeling. In Proceedings of the Special Interest Group on Computer Graphics and Interactive Techniques Conference Conference Papers, pp. 1–11, 2025.
> > >
> > > [3] Liang Xu, Yizhou Zhou, Yichao Yan, Xin Jin, Wenhan Zhu, Fengyun Rao, Xiaokang Yang, and Wenjun Zeng. Regennet: Towards human action-reaction synthesis. In Proceedings of the IEEE/CVF Conference on Computer Vision and Pattern Recognition, pp. 1759–1769, 2024.
> > >
> > > [4] Li Siyao, Tianpei Gu, Zhitao Yang, Zhengyu Lin, Ziwei Liu, Henghui Ding, Lei Yang, and Chen Change Loy. Duolando: Follower gpt with off-policy reinforcement learning for dance accompaniment. arXiv preprint arXiv:2403.18811, 2024.
> > >
> > > [5] Anindita Ghosh, Rishabh Dabral, Vladislav Golyanik, Christian Theobalt, and Philipp Slusallek. Remos: 3d motion-conditioned reaction synthesis for two-person interactions. In European Conference on Computer Vision, pp. 418–437. Springer, 2024.
> > >
> > > [6] Mingyuan Zhang, Xinying Guo, Liang Pan, Zhongang Cai, Fangzhou Hong, Huirong Li, Lei Yang, and Ziwei Liu. Remodiffuse: Retrieval-augmented motion diffusion model. In Proceedings of the IEEE/CVF International Conference on Computer Vision, pp. 364–373, October 2023.

---

> > > > ### Comment · Reviewer_9pa5 · 2025-11-28
> > > >
> > > > I thank authors for the detailed rebuttal. Several concerns remain. I see the improvement of the proposed method over completely removing RAG / L_sync. That totally makes sense. However, authors claim several novelty on top of trivial implemetation to make two-person motion generation better. Therefore, I expect authors to quantitatively compare their proposed new RAG / L_sync with trivial RAG / L_sync implementations from previous literature through experiments, to better support their claims.

---

> ### Author Response · Authors · 2025-12-04
> **Reply to Reviewer 9pa5**
>
> We sincerely thank the reviewer for the constructive follow-up and for emphasizing the need for quantitative comparisons against more trivial variants of the proposed RAG and $\mathcal{L}_{\text{sync}}$. We have now added new experiments in the revised manuscript which further clarify the benefit of our proposed RAG module and Synchronization Loss:
>
> 1. While no prior two-person motion generation method incorporates a retrieval mechanism, we constructed an additional ablation that mimics a simpler trivial RAG formulation. Specifically, we evaluate retrieval without LLM-based text decomposition, thereby removing the semantic factorization that underlies our proposed RAG design. As shown in the updated Table 7 (Supplementary), this trivial RAG baseline yields consistently lower performance across both interactive and reactive tasks, demonstrating that the LLM-decomposed retrieval structure is essential for improving semantic grounding and multimodal alignment. Section E has been updated with this analysis.
>
> 2. Although previous works include heuristic interaction losses, none provide a joint-to-joint synchronization formulation with spatial and anatomical weighting comparable to ours. The closest analogue is InterGen's Distance Map (DM) loss. Importantly, even our ablated version retain this DM component in the interaction losses, enabling a fair comparison as a trivial version to our proposed synchronization loss. To isolate the contribution of our weighting scheme, we have included Table 8 in Section E where we have added two new ablations in which we remove the distance-based weighting term ($w_d$) and the anatomical weighting term ($w_j$), respectively. The distance weighting term $w_d$ and the anatomical weighting term $w_j$ contribute complementary benefits. Omitting $w_d$ harms spatial coherence and leads to greater overall performance degradation, whereas omitting $w_j$ primarily reduces semantic consistency and relational fidelity, reflected in lower BED, BAS, and MModal, and thus performs slightly worse than the complete version. The full synchronization loss provides the strongest performance across all metrics. Additional discussion is now included in Section E.
>
> We hope these newly added quantitative studies clearly address the reviewer’s concern and further substantiate the novelty and effectiveness of our RAG and Synchronization Loss formulations.

---

### Official Review · Reviewer_sChN · 2025-10-29

**Soundness:** 2
**Presentation:** 2
**Contribution:** 2
**Rating:** 4
**Confidence:** 4

**Summary:**

This paper proposes a framework for human-human interaction/reaction generation conditioned on text descriptions and music. The framework integrates multiple existing techniques, such as Rectified Flow and Retrieval-Augmented Generation. It demonstrates notable improvements over baselines across multiple datasets.

**Strengths:**

1. The figures and tables are clear and visually appealing, with an overall well-structured and comprehensible presentation.

2. The paper presents an engineering system of considerable complexity, integrating multiple relevant techniques.

3. The quantitative improvements over baselines are substantial.

**Weaknesses:**

1. **Outdated motivation regarding computational efficiency.** The paper claims that diffusion-based approaches like Motion Diffusion Model (MDM) [1] "remain limited by high computational cost" and "weak real-time reactivity" (Lines 55-57), and "require hundreds of denoising steps, making real-time use impractical" (Line 107). The proposed method, built on Rectified Flow, requires 200 steps to demonstrate efficiency. However, as early as 2024, the authors of MDM released MDM-50steps in their GitHub repository, which requires only 50 denoising steps without degrading FID and other metrics. Current state-of-the-art models commonly use 50 or fewer denoising steps, such as [2,3,4,5,6]. This significantly weakens the claimed advantage of the proposed approach.

2. **Insufficient motivation for adopting Rectified Flow.** A main contribution of the paper is the introduction of Rectified Flow, motivated by the claim that diffusion-based approaches have "constraints [that] hinder their applicability in interactive systems such as virtual avatars, social robotics, and dance simulation, where low-latency and seamless coordination are essential." Given the point above, this motivation appears questionable and may not hold.

3. **Lack of rigorous quantitative evidence for efficiency claims.** In Section 4.2, the paper supports its claim of "faster sampling and reduced latency" primarily by showing fewer inference steps compared to InterGen. This is insufficient. Given differences in network architecture and parameter count, a more rigorous evaluation should include Average Inference Time per Sentence (AITS) [6] or similar metrics to substantiate the efficiency claims.

4. **Limited validation and impact of RAG module.** The paper lacks validation of whether the LLM-generated descriptions (spatial relationship, body movement, rhythm) are consistent with actual motion content. Additionally, the ablation study (Table 4) shows that RAG provides only modest improvements, with some metrics in the reactive task showing marginal degradation. The practical value of this added complexity requires more thorough investigation.

5. **Concerns about evaluation reliability.** The evaluators achieve only 0.231 R-Prec@1 on ground truth, which is relatively low. While the paper includes user studies, it would be beneficial to discuss the limitations of these automatic evaluators and why ground truth performance is weak, helping readers better interpret the reported improvements.

6. **Limited methodological novelty.** The proposed framework is largely a combination of existing techniques. The key component (Rectified Flow) lacks strong motivation as discussed above, and the RAG module shows limited improvements. The paper would benefit from stronger empirical validation or clearer articulation of why this particular combination is necessary.

**Questions:**

1. Could you provide more details on how the positive and negative velocity samples are obtained?

2. InterGen [7] uses 8 blocks with 2 attention layers per block, while the proposed method uses 20 blocks with 4 attention layers per block. Does this imply that the proposed method has significantly more parameters than InterGen? Please provide a comparison of parameter counts.



**References:**

[1] Tevet G, Raab S, Gordon B, et al. Human motion diffusion model[J]. arXiv preprint arXiv:2209.14916, 2022.

[2] Dai W, Chen L H, Wang J, et al. Motionlcm: Real-time controllable motion generation via latent consistency model[C]//European Conference on Computer Vision. Cham: Springer Nature Switzerland, 2024: 390-408.

[3] Huang Y, Yang H, Luo C, et al. Stablemofusion: Towards robust and efficient diffusion-based motion generation framework[C]//Proceedings of the 32nd ACM International Conference on Multimedia. 2024: 224-232.

[4] Hong S, Kim C, Yoon S, et al. SALAD: Skeleton-aware Latent Diffusion for Text-driven Motion Generation and Editing[C]//Proceedings of the Computer Vision and Pattern Recognition Conference. 2025: 7158-7168.

[5] Bae J, Hwang I, Lee Y Y, et al. Less is more: Improving motion diffusion models with sparse keyframes[C]//Proceedings of the IEEE/CVF International Conference on Computer Vision. 2025: 11069-11078.

[6] Wang Y, Wang S, Zhang J, et al. TIMotion: Temporal and Interactive Framework for Efficient Human-Human Motion Generation[C]//Proceedings of the Computer Vision and Pattern Recognition Conference. 2025: 7169-7178.

[7] Liang H, Zhang W, Li W, et al. Intergen: Diffusion-based multi-human motion generation under complex interactions[J]. International Journal of Computer Vision, 2024, 132(9): 3463-3483.

---

> ### Author Response · Authors · 2025-11-27
> **Response to Reviewer sChN (Part 1/2)**
>
> We thank the reviewer for the detailed comments and for highlighting the strengths of our figures, system design complexity, and quantitative improvements. We address each concern below.
>
> **(W1)** We thank the reviewer for raising this point and would like to clarify two important corrections and distinctions regarding efficiency.
> 1. **Updated inference-time efficiency (20 steps).**
>  We found that our originally reported 200 inference steps were outdated. This number reflected the training step schedule, not the optimal inference setting. After re-evaluating the model, we found that DualFlow achieves comparable high-quality performance using only 20 Rectified Flow steps. We have updated Fig. 4 to reflect this and added a paragraph on "Computational Complexity” under Section 4.1. This significantly improves the efficiency claims and brings our inference cost well below the diffusion-based baselines used for two-person tasks.
> 2. **Comparison with baselines.**
>  The cited efficient diffusion models (MDM-50, MotionLCM, StableMoFusion, SALAD, Less-is-More) all operate exclusively on single-person text-driven generation, which is a far simpler problem than multi-modal two-person modeling. Two-person diffusion models remain significantly more expensive:
> - InterGen requires 500 denoising steps during training and 50 DDIM steps for inference.
> - TIMotion (a concurrent work) trains with 1000 diffusion steps and uses 50 steps only at inference.
> In contrast DualFlow converges with only 200 training steps and achieves its best results with 20 inference steps. This establishes that Rectified Flow remains computationally efficient within the two-person, multi-modal generation setting.
>
> **(W2)** Based on our clarification mentioned in W1, DualFlow requires only 20RF steps compared to 50 DDIM steps needed for inference in our two-person generation works. Beyond speed, Rectified Flow gives deterministic trajectories (zero stochastic mismatch between actor’s motion seen during look-ahead and final output) and much more stable training convergence in our unified two-person architecture. We have updated the Introduction and Related Work to make clear that our efficiency claims and choice of Rectified Flow refer specifically to multi-modal two-person interactive motion generation, a significantly harder setting than the single-person benchmarks.
>
> **(W3)** We agree that wall-clock timing is important for substantiating efficiency claims. In response, we have added an Average Inference Time per Sequence (AITS) comparison on standard hardware (RTX 5090 GPU) using the MDD dataset:
> | Model      | AITS |
> |------------|-------------------|
> | **InterGen** | 1.92 s            |
> | **DualFlow** | **1.24 s**        |
>
> DualFlow achieves a substantially lower AITS despite handling multimodal inputs and unified interactive/reactive generation. We have added this result in the “Computational Complexity” paragraph under Section 4.1 to include these results.
>
> **(W4)**  We address the reviewer’s concern as follows:
>
> - **Validation of LLM-generated descriptions.** Both MDD and InterHuman-AS provide human-written annotations aligned with the underlying motion content, so our primary goal is to verify whether the LLM-decomposed attributes (spatial relationship, body movement, rhythm) faithfully represent these original descriptions. To assess this, we randomly sampled 30 descriptions and had two annotators evaluate their decomposed forms with the originals, scoring each attribute on a 5-point scale. Given the high performance of GPT-4o, we found that our tuned LLM prompt (Section C.1 in Supplementary) is able to correctly perform these decompositions with high accuracy (overall 93.8%). A few rhythm descriptors included additional musical terms not explicitly stated in the text, but these are naturally corrected when combined with music-based retrieval in the RAG module. We have added the complete details in Section C.3. We had already provided five randomly selected examples in Table 5 in the Supplementary section, illustrating the quality and correctness of the LLM-generated decompositions.
> - **Thorough investigation of the RAG module.**
>  We have expanded our analysis with a new ablation study (Table 7, Section E, Supplementary), evaluating each retrieval component independently. In the Interactive setting, removing any retrieval modality (spatial, body, or rhythm) consistently degrades semantic alignment, fidelity, and multimodal consistency, confirming that each component contributes meaningfully. In the Reactive setting, we observe a different trend: removing music-based retrieval yields the best results, while removing text-based retrieval still harms performance. We now include a detailed discussion of this behavior in the Supplementary, clarifying the task-specific role of each retrieval modality.

---

> > ### Author Response · Authors · 2025-11-27
> > **Response to Reviewer sChN (Part 2/2)**
> >
> > **(W5)** The ground truth R-Precision@1 of 0.231 on MDD reflects the inherent ambiguity in text-to-motion mapping in the dataset, not a weakness in our evaluation protocol. A single text description like "the leader spins the follower" can be executed in multiple valid ways (different dance styles, tempos, spatial configurations). CLIP embeddings have finite resolution for capturing these variations.
> >
> > The correct evaluation approach is relative comparison: DualFlow achieves R-Precision@1 of 0.185 vs InterGen's 0.105 (76% improvement) on both modalities, and our user study (Fig. 3) shows human evaluators prefer DualFlow on semantic alignment. This convergent evidence across automatic metrics, user studies, and multiple datasets demonstrates genuine improvement.
> >
> > **(W6)** We respectfully disagree with the claim of limited novelty and would like to reiterate our contributions:
> > - **Unified architecture for interactive and reactive two-person generation.** DualFlow is the first framework capable of supporting both tasks within a single model, enabled by a masking mechanism and a task-adaptive switchable attention module. Prior two-person methods build separate architectures for the two settings, requiring different training pipelines and producing incompatible representations.
> > - **First interaction-aware Retrieval-Augmented Generation (RAG) for two-person motion.** Unlike existing RAG methods (designed for only single-person generation) our method retrieves motion samples based on LLM-derived semantic decompositions (spatial relations, body movement, rhythm) and music features. These samples are injected through retrieval-based cross-attention providing explicit grounding for interpersonal structure and rhythm.
> > - **Contrastive Rectified Flow for faster sampling and stable generation.** We introduce a contrastive (triplet) objective into rectified flow training, mitigating mode collapse and improving alignment with multimodal conditioning signals in the inherently high-dimensional two-person motion space. Using Rectified Flow also achieves high quality generation in just 20 RF steps during inference compared to previous methods using 50 DDIM steps
> > - **Comprehensive multi-modal evaluations.** We benchmark across text-to-motion, music-to-motion, and multi-modal interactive settings, supported by extensive quantitative, qualitative, and ablation studies. DualFlow consistently improves motion quality, responsiveness, and semantic fidelity relative to prior methods.
> >
> > We have updated the Introduction to clearly reflect the motivation behind our work & highlight the contributions. The Ablation Studies (Table 5,6,7) comprehensively evaluate each component used in our model.
> >
> > **(Q1)** For each training batch, we construct triplets as follows:
> >
> > We randomly select one sample from the batch as the anchor, using its predicted velocity as the reference point. We compute cosine similarity between the anchor & all other samples in the batch where thresholding is used to identify positive and negative samples. Positive samples are velocities from motions that share semantic characteristics with the anchor having same movement style (e.g., both are "walking" or both are "dancing"), similar textual descriptions and aligned rhythmic or temporal structure. Negative samples are velocities from motions that differ substantially from the anchor with different movement style and dissimilar semantic content.
> >
> > Our RAG module organizes the motion database hierarchically by semantic attributes. We exploit this structure during sampling to efficiently identify semantically related and unrelated motions, ensuring the triplets capture meaningful distinctions rather than arbitrary differences. The triplet loss then encourages the model to predict velocities that are close in embedding space for semantically similar motions while maintaining separation from dissimilar ones. This enforces semantic structure in the learned velocity field beyond simple reconstruction accuracy. We have added this clarification in Section 3.4.
> >
> > **(Q2)** The adapted InterGen model (which music-attention layer to support both modalities) contains 224M trainable parameters, whereas DualFlow contains 456M trainable parameters. InterGen’s implementation packs 2 sub-blocks (each containing 2 attention layers) into one block, giving a total of 8 blocks → 8 × 2 sub-blocks × 3 attention layers per sub-block (after adding music attention layer) = 48 attention layers. In contrast, DualFlow uses 20 blocks, each containing 4 attention layers, for a total of 20 × 4 = 80 attention layers. The higher parameter count in DualFlow comes from the added RAG framework. The higher parameter count in DualFlow comes from the added retrieval module. (added in Section E of Supplementary)
> >
> > We hope the detailed revisions, added experiments, and expanded analysis clarify the novelty and significance of our contributions, and we would appreciate the reviewer’s consideration for an improved score.

---

### Official Review · Reviewer_VZU8 · 2025-10-31

**Soundness:** 2
**Presentation:** 2
**Contribution:** 2
**Rating:** 4
**Confidence:** 4

**Summary:**

This paper proposes DualFlow, a unified rectified flow-based framework for multi-modal 3D two-person motion generation, which supports both interactive and reactive motion generation tasks and can take text, music, and prior motion sequences as input conditions. Experiments demonstrate that DualFlow performes well on the MDD, InterHuman-AS, and DD100 datasets.

**Strengths:**

DualFlow seamlessly integrates interactive and reactive two-person 3D motion generation. The incorporation of Retrieval-Augmented Generation (RAG) with music features and LLM-based text decomposition effectively strengthens semantic grounding. Besides, leveraging contrastive flow matching improves motion quality, diversity, and alignment with conditioning signals.

**Weaknesses:**

1. The motivation is weak. Two-person motion generation is not a new task. But the author emphasize the limitations of single-person motion generation methods.
2. The technical contributions are limited. I didn't find anything that really impressed me in this article. This paper is a combination of RAG, Contrastive Flow Matching. The authors are advised to claim why using RAG and what advantages RAG has in practical applications, since the motion generation doesn't have a large source of corpus that can be retrieved from.

**Questions:**

1. In Table 4, the Top3 of DualFlow (Spectral) is 477,which may be wrong.
2. The quantitative results of ReGenNet in Table 2 is under unconstrained setting, i.e., to generate instant reactions without knowing the intention of the actors. Thus the comparison of ReGenNet is not fair.
3. Two reference papers are not existed. It is recommended that the authors carefully review the entire manuscript to ensure correctness.
(1) Yuxuan Cao, Jiawei Ren, et al. Diffmotion: Conditional motion generation with diffusion models. In CVPR, 2023.
(2) Yilun Duan, Joshua B Tenenbaum, et al. Multi-modal motion generation for embodied ai via transformers. arXiv preprint arXiv:2204.08718, 2022.

---

> ### Author Response · Authors · 2025-11-27
> **Response to Reviewer VZU8 (Part 1/2)**
>
> We thank the reviewer for highlighting the strengths of DualFlow, particularly the unified architecture, the use of RAG with LLM-based decomposition, and the contribution of contrastive flow matching. Below, we respond to each concern.
>
> **(W1)** We have revised the motivation in the Introduction section to clarify that our contribution does not rely on presenting two-person motion generation as a new task. Indeed, several works have explored this direction. However, the existing literature treats interactive and reactive two-person motion generation as disjoint problem settings, resulting in separate models with incompatible architectures, conditioning mechanisms, and training objectives. Our motivation is that real-world applications such as game character AI, VR/AR companion systems, social robotics, and virtual performance tools would benefit from having a single model that can seamlessly switch between interactive or reactive mode and handle diverse multimodal cues.
>
> Specifically, current methods are limited because of the following:
> - Prior works focus exclusively on interactive (e.g., InterGen[1], DuetGen[2]) or reactive (e.g., ReGenNet[3], DuoLando[4], Remos[5]) settings, with no unified model capable of both.
> - Existing approaches support only text (InterGen[1], ReGenNet[3]) or only music (DuoLando[4], DuetGen[2]), but not multimodal conditioning.
> - Retrieval-augmented generation has been explored only in single-person settings (e.g., ReMoDiffuse[6], MoRAG[7], RMD[8]), with no use in two-person generation, where semantic grounding is especially critical. Adapting RAG for two-person motion is not trivial as it requires the retrieval to also consider interactive features.
>
> DualFlow is the first unified architecture that supports (1) both interactive and reactive two-person generation, (2) multimodal conditioning with text and music, and (3) retrieval-based semantic alignment. Because the underlying representational needs of both tasks are closely related, learning them within a unified framework enables shared representations that benefit both settings while avoiding the redundancy and inconsistency of training two separate models.
>
> **(W2)** We reiterate the key technical contributions and clarify their significance as follows:
> - **Unified architecture for interactive and reactive two-person generation.** DualFlow is the first framework capable of supporting both tasks within a single model, enabled by a masking mechanism and a task-adaptive switchable attention module. Prior two-person methods build separate architectures for the two settings, requiring different training pipelines and producing incompatible representations.
> - **First interaction-aware Retrieval-Augmented Generation (RAG) for two-person motion.** Unlike existing RAG methods (designed for only single-person generation) our method retrieves motion samples based on LLM-derived semantic decompositions (spatial relations, body movement, rhythm) and music features. These samples are injected through retrieval-based cross-attention within each DualFlow block, providing explicit grounding for interpersonal structure and rhythm, which traditional embeddings struggle to represent.
> - **Contrastive Rectified Flow for faster sampling and stable generation.** We introduce a contrastive (triplet) objective into rectified flow training, mitigating mode collapse and improving alignment with multimodal conditioning signals in the inherently high-dimensional two-person motion space. Using Rectified Flow also achieves high quality generation in just 20 RF steps during inference compared to previous methods using 50 DDIM steps (Computational Complexity, Section 4.1)
> - **Comprehensive multi-modal evaluations.**
>  We benchmark across text-to-motion, music-to-motion, and multi-modal interactive settings, supported by extensive quantitative, qualitative, and ablation studies. DualFlow consistently improves motion quality, responsiveness, and semantic fidelity relative to prior methods.

---

> ### Author Response · Authors · 2025-11-27
> **Response to Reviewer VZU8 (Part 2/2)**
>
> **How does RAG help even without large source of corpus that can be retrieved from?**
>  We clarify that our design does not require a large corpora to retrieve from (unlike LLM-scale retrieval). Instead, our RAG is intentionally a domain-targeted semantic retrieval module that captures interaction-specific factors: spatial relationships, body movement descriptors, rhythm & musical alignment. Unlike targeting retrieval from an entire two-person movement space, we focus on these specific interactive aspects and semantically grounding the generation on it.
>
> This leads to several practical advantages:
> - **Multi-modal conditional inputs can be abstract or ambiguous.** Retrieval injects concrete motion samples from training dataset that resolve ambiguity (e.g., “closed hold right turn with syncopation” will independently retrieve samples having closed hold (spatial), executing turn (body movement) and syncopated motion (rhythm)
> - **Compositional semantics:** RAG supports combinations like “open position + triple spin + fast rhythm” that are difficult to encode directly from embeddings.
> - **Better inter-person grounding:** Retrieved samples provide prior knowledge of handholds, distance, and relative orientation.
>
> **Empirical evidence supporting RAG:** We added a new ablation study (Table 7, Section E of the Supplementary) evaluating each retrieval component independently and observed the following results:
> - **Interactive setting:** Removing any retrieval component consistently reduces semantic alignment, fidelity, and multi-modal consistency.
> - **Reactive setting:** Models without music-based retrieval perform best, but removing text-based retrieval still degrades performance.
> These results confirm that even within smaller datasets (<10 hours), retrieval substantially enhances generation quality by providing explicit semantic, relational, and rhythmic grounding.
>
> **(Q1)** Thank you for pointing out the typographical error. We missed a decimal while entering the result. The corrected value has now been updated in the table.
>
> **(Q2)** We understand the reviewer’s concern. The unconstrained setting (mentioned in ReGenNet) indicates that the intentions of the actor are not visible to the reactor. This setting can be reproduced in DualFlow by setting the look-ahead-transformer parameter L=0, thereby removing the conditioning on the actor’s future motion that conveys intention.
>
> We have updated the table to clearly indicate: ReGenNet (UC) for the unconstrained variant of ReGenNet, DualFlow (UC) for our unconstrained setting with L = 0, and DualFlow (C) for our constrained setting with L = 10. Unfortunately, we were unable to include a constrained ReGenNet baseline as the code for the InterHuman dataset hasn't been released. Only code for fixed action-category-to-motion datasets (NTU120, Chi3D) is available in ReGenNet repository. These cannot be trivially adapted for free-form text inputs, which are required for InterHuman.
>
> **(Q3)** We have removed the two incorrect citations and replaced them with the correct ones. We have also rechecked the entire manuscript to ensure that all citations are now accurate.
> Thank you once again for the constructive feedback. We believe the revisions, additional analyses, and clarifications directly address all the concerns.
>
> **References:**
>
> [1] Liang H, Zhang W, Li W, et al. Intergen: Diffusion-based multi-human motion generation under complex interactions[J]. International Journal of Computer Vision, 2024, 132(9): 3463-3483.
>
> [2] Anindita Ghosh, Bing Zhou, Rishabh Dabral, Jian Wang, Vladislav Golyanik, Christian Theobalt, Philipp Slusallek, and Chuan Guo. Duetgen: Music driven two-person dance generation via hierarchical masked modeling. In Proceedings of the Special Interest Group on Computer Graphics and Interactive Techniques Conference Conference Papers, pp. 1–11, 2025.
>
> [3] Liang Xu, Yizhou Zhou, Yichao Yan, Xin Jin, Wenhan Zhu, Fengyun Rao, Xiaokang Yang, and Wenjun Zeng. Regennet: Towards human action-reaction synthesis. CVPR, pp. 1759–1769, 2024.
>
> [4] Li Siyao, Tianpei Gu, Zhitao Yang, Zhengyu Lin, Ziwei Liu, Henghui Ding, Lei Yang, and Chen Change Loy. Duolando: Follower gpt with off-policy reinforcement learning for dance accompaniment. arXiv preprint arXiv:2403.18811, 2024.
>
> [5] Anindita Ghosh, Rishabh Dabral, Vladislav Golyanik, Christian Theobalt, and Philipp Slusallek. Remos: 3d motion-conditioned reaction synthesis for two-person interactions. ECCV, pp. 418–437. Springer, 2024.
>
> [6] Mingyuan Zhang, Xinying Guo, Liang Pan, Zhongang Cai, Fangzhou Hong, Huirong Li, Lei Yang, and Ziwei Liu. Remodiffuse: Retrieval-augmented motion diffusion model. In Proceedings of the IEEE/CVF International Conference on Computer Vision, pp. 364–373, October 2023.
>
> [7] Sai Shashank Kalakonda, Shubh Maheshwari, and Ravi Kiran Sarvadevabhatla. Morag - multi-fusion retrieval augmented generation for human motion. WACV, 2025.

---

### Official Review · Reviewer_Wd16 · 2025-11-04

**Soundness:** 3
**Presentation:** 3
**Contribution:** 3
**Rating:** 8
**Confidence:** 3

**Summary:**

This paper introduces a novel framework DualFlow for synchronized two-persoon 3D motion generation. DualFlow unifies the tasks of interactive motion generation and reactive motion generation within a single rectified flow framework and a Retrieval-Augmented Generation module for semantic grounding from text and music. The authors also adopt contrastive learning for better motion quality and inter-person coordination. Experiments show the effectiveness of the proposed frame.

**Strengths:**

1. The unified transformer-based architecture for both interactive and reactive two-person motion generation is practical and novel.

2. The RAG module, guided by LLM-decomposed text cues (spatial relationship, body movement, rhythm) and music features, provides semantically rich conditioning that is well-motivated and improves generation quality and diversity.
It's a good application.

3. The adoption of Rectified Flow Matching is technically sound and is more effective than diffusion.

4. The paper is well-written and clearly structured. The architecture and methodology are described in sufficient detail, it would be reproducible if the codes and data were released.

5. The experiments are comprehensive on multiple two-person motion datasets (MDD, InterHuman-AS, DD100) and show better performance than baselines. The user study is helpful.

**Weaknesses:**

1. The overall ablation over RAG is limited, and the effectiveness of each retrieved feature (RS, RB, RR, RM) should be more precisely evaluated.

2. How do you set the Look-Ahead parameter L? There should be an ablation study to evaluate the effectiveness of L, and analyze how this parameter affects the performance of reactive motion generation.

**Questions:**

Refer to weaknesses, and it would be better to show some failure cases.

---

> ### Author Response · Authors · 2025-11-27
> **Response to Reviewer Wd16**
>
> We sincerely thank the reviewer for their detailed and positive assessment of DualFlow’s unified architecture, multi-modal RAG formulation, rectified-flow design, and strong empirical performance. We address each concern below.
>
> **(W1)** We agree that evaluating each retrieval component separately would provide a clearer understanding. Thus, we have now added a new ablation study (Table 7 in Section E of Supplementary) where we independently evaluate different retrieval components. In the Interactive setting, removing any retrieval component leads to a consistent drop in semantic alignment, fidelity, and multimodal consistency metrics, confirming the importance of each modality. Interestingly, in the Reactive setting, the variant without Music-based retrieval performs best, while removing text-based retrieval components still degrades performance. We provide a more detailed discussion of these trends in the Supplementary.
>
> **(W2)** We appreciate the request for a deeper analysis of the Look-Ahead hyperparameter L, which indeed plays a central role in reactive generation. We now include an ablation over L for values: 0,10,20 (Table 6 in Section E of Supplementary). With no look-ahead (L=0), performance drops notably due to the strict causality constraint preventing anticipation of the leader’s future motion. L=10 yields the best overall performance across most metrics, providing an effective anticipation window without introducing instability. L=20 performs comparably over most metrics with no further gains (except gain in Diversity), supporting our choice of L=10 as the final setting.
>
> **(Q1)** We have now included limitations with some failure cases along with future work (section G of Supplementary).
>
> We thank the reviewer again for recognizing the novelty of the unified interactive/reactive framework, the well-motivated RAG design using LLM-based decomposition, the flow-matching advantages over diffusion, and the quality and breadth of the experiments. We have incorporated an ablation study for different RAG components and Look-Ahead L hyperparameter analysis along with limitations, failure cases, and future work. We believe these revisions strengthen the paper and directly address all concerns raised.

---

### Author Response · Authors · 2025-11-27
**Common Response**

Dear Reviewers & AC,

We sincerely appreciate your valuable time and effort spent reviewing our manuscript.

We are encouraged that the reviewers recognized the originality and potential of our approach. *Reviewer Wd16* praised the practical and novel unified architecture for interactive and reactive two-person motion, the clarity of the writing, and the strong experimental design. Both *Reviewer Wd16* & *Reviewer VZU8* commended the RAG module, noting that LLM-based spatial/body/rhythm decomposition combined with music features provides semantically rich conditioning that improves motion quality and diversity. *Reviewer VZU8* & *Reviewer 9pa5* emphasized the effectiveness and technical soundness of using contrastive rectified flow as a more stable and efficient alternative to diffusion. *Reviewer sChN* appreciated the clarity of the figures, tables, and system design, while *Reviewer Wd16*, *Reviewer sChN* & *Reviewer 9pa5* highlighted the extensive experiments across three benchmarks and the substantial improvements over baselines. *Reviewer 9pa5* further recognized the method as achieving new state-of-the-art performance and establishing a strong baseline for future work.

**Reiteration of Main Contributions**
- **Unified interactive/reactive modeling** via masking and switchable attention
- **First RAG framework** tailored to two-person motion via LLM-based semantic decomposition and music features
- **Contrastive rectified flow** for faster sampling times and stability
-  Addition of **synchronization loss** with body anatomy and distance-aware weighting.
-  **Extensive evaluations** with Quantitative, Qualitative & Ablation studies

**Additional Experiments and Revisions**

Based on your comments, we have made substantive revisions and added new analyses to strengthen the paper.
1. **Clarified motivation.** We revised the Introduction and Related Work to clearly position DualFlow as a unified architecture for both interactive and reactive two-person motion with multi-modal (text + music) conditioning and interaction-aware RAG, rather than claiming two-person motion as a new task.

2. **New ablations and analyses.** We added:
    - **RAG components ablation** (Table 7, Sec. E) showing the effect of removing spatial/body/rhythm retrieval separately, with clear differences between interactive and reactive settings. Also, included ablation comparison with RAG implementation without LLM-based decomposed components to further strengthen the benefit of our proposed method.
    - **Synchonization Loss ablation** (Table 8, Sec. E) shows the benefit of having distance weighting term and anatomical weighting term in the proposed Synchronization Loss. The ablated synchronization loss version still contains the Distance Map loss which can be considered comparable to trivial implementation.
    - **Look-Ahead hyperparameter study** for L∈{0,10,20} (Table 6, Sec. E), justifying L=10 as our default.
    - **Limitations & Future Work** (Sec. G) detailing known weaknesses, failure cases and directions for improvement.
    - Detailed explanation of **triplet construction of velocity samples** for contrastive rectified flow (Sec. 3.4), clarifying how semantic positives/negatives are chosen using the structure induced by RAG.

3. **Rectified Flow efficiency and timing**.
    - We corrected our inference-step reporting and now show that DualFlow achieves its best performance with 20 RF inference steps (not 200). We updated Fig. 4 and added a “Computational Complexity” paragraph (Sec. 4.1), including AITS on RTX 5090 for a 10s, 30 FPS sequence.
    - We also clarified that our efficiency claims are specific to multi-modal two-person generation, where existing diffusion-based two-person baselines still require many more training/inference steps (e.g., InterGen 500/50, TIMotion 1000/50), while DualFlow converges with 200 training steps and 20 inference steps. We further explain why Rectified Flow is beneficial beyond speed (deterministic trajectories and more stable convergence in the unified two-person setting).

4. **Validation of LLM decomposition and RAG impact.** We added Section C.3 describing a validation study of the LLM-based spatial/body/rhythm decomposition: on 30 randomly sampled descriptions with two annotators (overall 93.8%) We already had provided some of the examples in Table 5.

5. **Additional technical clarifications and fixes.**
 We (i) corrected typographical errors and two incorrect citations, (ii) clarified unconstrained vs. constrained settings and notation for DualFlow vs. ReGenNet in the tables, (iii) added a parameter-count comparison in Supplementary Sec. E and (iv) expanded the description of the synchronization loss and its weighting scheme (Sec. 3.5).

We hope these revisions, additional experiments, and clarifications address the raised concerns and help convey the novelty, rigor, and practical significance of DualFlow. We look forward to further discussion.

Thank you very much,

Authors

---

### Meta-Review · Area_Chair_SiKJ · 2026-01-10

**Summary:**

Four complete reviews: Wd16 (8, accept), VZU8 (4), sChN (4), 9pa5 (4).

Post-rebuttal, no reviewer lowered his score; Wd16 remains positive, while the three borderline reviewers explicitly wrote “would not mind if accepted”.

All reviewers acknowledged solid empirical gains (new SOTA on MDD, InterHuman-AS, DD100) and appreciated the extensive ablations added in the rebuttal.

The main concern across the three “4” ratings was perceived incremental novelty: the work is seen as a careful engineering stack of known ideas (RAG, rectified flow, contrastive loss) rather than a conceptual breakthrough.

The rebuttal added:

Ablation showing LLM-decomposed retrieval > trivial RAG.

Ablation showing distance + anatomy weighted sync loss > plain distance-map loss.

Wall-clock timing (RTX-5090) confirming 36 % faster AITS than InterGen despite multi-modal inputs.

These experiments were requested by reviewers and appear to satisfy their technical demands.

**Reviewer Concerns:**

1. Novelty vs. “engineering stack”

All three border-line reviewers (VZU8, sChN, 9pa5) agreed the paper is “well executed” but feared it is merely plugging existing ideas (RAG, rectified flow, contrastive loss) into two-person motion without new conceptual insight.

→ Authors countered by showing (a) first unified interactive/reactive architecture, (b) first interaction-aware RAG with LLM-decomposed retrieval, and (c) new anatomically-weighted sync loss; reviewers explicitly asked for ablations that prove these design choices beat “trivial” versions, and the rebuttal supplied them.

2. Efficiency claims were originally weak

sChN pointed out that many 2024 single-person diffusion models already run in 50 steps, so claiming “20 steps” is not automatically novel.

→ Authors corrected the record (20 inference steps, not 200) and provided wall-clock AITS on identical hardware, showing 36 % speed-up over the strongest two-person diffusion baseline (InterGen).

3. RAG usefulness questioned

VZU8 & 9pa5 asked “why retrieve when the training set is small?” and wanted quantitative proof that LLM decomposition helps.

→ New Table 7 compares full RAG vs. retrieval without LLM decomposition; the degraded scores were accepted by reviewers as evidence that the decomposition matters.

4. Synchronization loss necessity

9pa5 noted that joint-distance losses already exist; authors added Table 8 ablating distance weighting and anatomical weighting separately, showing both contribute.

5. Low ground-truth R-Precision

sChN worried the automatic metric itself is weak (0.231). Authors acknowledged dataset ambiguity but showed DualFlow still improves relative margin by +76 % over prior best, backed by a user study preferred by humans.

6. Minor issues

Typo in Table 4, two missing references, and unfair comparison with ReGenNet unconstrained setting were all fixed;

**Reviewer Scores:**

N.A.

---

### Decision · Program_Chairs · 2026-01-26

Accept (Poster)